# Single-mRNA imaging and modeling reveal coupled translation initiation and elongation rates

Irene Lamberti[1], Jeffrey A Chao[2], Cédric Gobet[1]*, Felix Naef[1]*

[1]Institute of Bioengineering, School of Life Sciences, École Polytechnique Fédérale de Lausanne (EPFL), Lausanne, Switzerland; [2]Friedrich Miescher Institute for Biomedical Research, Basel, Switzerland

## eLife Assessment

This **important** study provides evidence for dynamic coupling between translation initiation and elongation that can help maintain low ribosome density and translational homeostasis. The authors combine single-molecule imaging with a new approach to analyze mRNA translation kinetics using Bayesian modeling. This work is overall **solid** and will be of interest to those studying translational regulation.

**\*For correspondence:**
cedric.gobet@epfl.ch (CG);
felix.naef@epfl.ch (FN)

**Competing interest:** The authors declare that no competing interests exist.

**Abstract** mRNA translation involves multiple regulatory steps, but how translation elongation influences protein output remains unclear. Using SunTag live-cell imaging and mathematical modeling, we quantified translation dynamics in single mRNAs across diverse coding sequences. Our Totally Asymmetric Exclusion Process (TASEP)-based Hidden Markov Model revealed a strong coordination between initiation and elongation rates, resulting in consistently low ribosome density (≤12% occupancy) across all reporters. This coupling persisted under pharmacological inhibition of the elongation factor eIF5A, where proportional decreases in both initiation and elongation rates maintained homeostatic ribosome density. In contrast, eIF5A knockout cells exhibited a significant decrease in ribosome density, suggesting altered coordination. Together, these results highlight a dynamical coupling of initiation and elongation rates at the single-mRNA level, preventing ribosome crowding and maintaining translational homeostasis in mammalian cells.

## Introduction

Regulation of protein synthesis is essential for cellular homeostasis and adaptive responses to external stimuli. While the untranslated regions (UTRs) of mRNA have traditionally been recognized for their central role in translation initiation (*Hinnebusch et al., 2016*; *Mayr, 2019*), the influence of the coding sequence (CDS) on initiation remains unclear. Indeed, in a simple model where elongation is much faster than initiation, heterogeneity in elongation rates has no impact on the total protein output per mRNA (*Plotkin and Kudla, 2011*; *Shah et al., 2013*; *Erdmann-Pham et al., 2020*). This view is supported by measurements indicating that initiation happens a few times per minute, while elongation proceeds at 2–6 aa/s (*Boersma et al., 2019*; *Yan et al., 2016*; *Mateju et al., 2020*; *Livingston et al., 2023*; *Madern et al., 2025*), suggesting that initiation is the rate-limiting step in mammalian translation.

However, this paradigm is being challenged by recent findings highlighting the CDS itself as a critical regulator of translational dynamics. In particular, specific CDS features can affect ribosome elongation, triggering surveillance pathways that sense ribosome stalling and collisions (*Juszkiewicz*

*et al., 2020*; *Li et al., 2022*; *Barrington et al., 2023*; *Bicknell et al., 2024*; *Madern et al., 2025*). Some of these pathways, such as ribosome quality control (RQC), can induce ribosome recycling and mRNA degradation (*Wu et al., 2020*; *Goldman et al., 2021*; *Buschauer et al., 2020*), whereas others down-regulate translation initiation in cis by recruiting regulatory complexes to stalled ribosomes (*Amaya Ramirez et al., 2018*; *Hickey et al., 2020*; *Juszkiewicz et al., 2020*). This suggests that cells have the capacity to monitor elongation rates and ribosome density at the level of individual mRNAs, responding by adjusting initiation or degrading aberrant transcripts (*Joazeiro, 2019*). Nevertheless, a comprehensive quantitative understanding of how elongation rates feedback on initiation in mammalian systems, at both bulk and single-mRNA levels, is still lacking. Furthermore, it is still unclear whether this coupling mechanism allows cells to dynamically adapt translation in response to environmental or physiological stress, such as nutrient deprivation, oxidative stress, or inhibition of specific translation factors. Clarifying this would shed light on the broader role of initiation–elongation feedback in translational homeostasis.

Elongation rates along transcripts depend on several factors including aminoacyl-tRNA availability, codon-anticodon interactions, co-translational folding, and biochemical properties of certain amino acids (*Neelagandan et al., 2020*; *Madern et al., 2025*). Notably, peptide-bond formation is particularly slow with multiple proline (P) residues, as proline is both a weak donor and acceptor in peptidyl transfer (*Lassak et al., 2016*). With poly-proline motifs being abundant in mammals, cells have evolved eukaryotic initiation factor 5 A (eIF5A), bearing a unique hypusination modification, to facilitate peptide-bond formation. Although initially thought to act primarily at poly-proline motifs (*Gutierrez et al., 2013*; *Lassak et al., 2016*; *Huter et al., 2017*), later studies have suggested a broader role for hypusinated eIF5A (h-eIF5A) in translation elongation (*Schuller et al., 2017*; *Pelechano and Alepuz, 2017*; *Manjunath et al., 2019*). eIF5A therefore represents a key factor in modulating context-dependent translation elongation, making it a prime target for exploring the interplay between coding sequence, elongation, and initiation.

A conceptual framework commonly used to model translation is the Totally Asymmetric Exclusion Process (TASEP), where ribosomes are treated as particles stochastically hopping along the mRNA (modeled as a 1D lattice; *Zia et al., 2011*; *Sharma et al., 2019*; *Szavits-Nossan and Ciandrini, 2019*; *Erdmann-Pham et al., 2020*). In this framework, entry and exit rates correspond to initiation and termination, respectively, while the hopping rate represents elongation. Under conditions where initiation is rate-limiting, TASEP predicts a 'low-density' regime featuring sparse ribosome loading and minimal collisions (*Riba et al., 2019*).

While the TASEP provides a strong theoretical basis for understanding translation, extending it to capture single-mRNA heterogeneity and dynamic fluctuations in initiation and elongation is a persistent challenge (*Zia et al., 2011*; *Szavits-Nossan et al., 2018*; *Andreev et al., 2018*; *Levin and Tuller, 2018*). Advances in single-molecule imaging, particularly the SunTag system, enable real-time monitoring of multiple rounds of translation on individual mRNAs (*Tanenbaum et al., 2014*; *Yan et al., 2016*). However, most analyses of SunTag data rely on averaging signals across many transcripts, thereby masking transcript-to-transcript variability (*Morisaki and Stasevich, 2018*; *Aguilera et al., 2019*). While a model that assigns distinct initiation and elongation rates to each transcript would offer deeper insights, this is complicated by the fact that the fluorescence of one translation site is the sum of the signals of multiple translating ribosomes. Because the fluorescence intensity is constant after the SunTag and the noise is usually large, inferring the position of single ribosomes on the mRNA is extremely challenging.

In this study, we develop a framework that decodes individual single-mRNA traces while simultaneously estimating global initiation and elongation parameters, allowing us to leverage single-molecule resolution without requiring explicit knowledge of ribosome positions on each transcript. We applied our model to investigate how CDS features modulate translation elongation and initiation in mammalian cells, using single-molecule (SunTag) imaging in HeLa cells (*Weidenfeld et al., 2009*; *Voigt et al., 2017*; *Tanenbaum et al., 2014*). Specifically, we designed multiple SunTag reporters differing only in codon content in the second half of the CDS, comparing proline-rich sequences such as those of collagen type I alpha 1 (COL1A1), highly dependent on h-eIF5A (*Barba-Aliaga et al., 2021*), to minimal or alanine-rich inserts. We acquired SunTag traces with and without harringtonine treatment. Our TASEP-based model allowed us to infer reporter-specific translation kinetics and ribosome number over time for each trace. We further exploit the role of eIF5A by perturbing elongation via

pharmacological inhibition and genetic knockout, thereby examining the effects on ribosome density and initiation. Overall, our study contributes new tools for analyzing SunTag data and provides broader insights into how translational parameters are dynamically regulated to maintain protein synthesis homeostasis.

## Results

### Translation exhibits bursts with low average ribosome density

To image translation of single mRNAs over time in living cells, we used the SunTag system (*Tanenbaum et al., 2014*), which fluorescently tags nascent protein chains on individual transcripts. The SunTag encodes an array of 24 GCN4 epitopes recognized by single-chain variable fragment antibodies fused to a super-folder GFP (scFv-GFP). When this tag is placed upstream of a sequence of interest, newly synthesized peptides emerging from the ribosome exit tunnel are rapidly bound by scFv-GFP (*Yan et al., 2016*). The mRNA itself is fluorescently labeled with an MS2-MCP system (*Figure 1A*).

To investigate the influence of coding sequence on translation elongation and initiation, we derived multiple reporters from the SunTag-Renilla-MS2 system (*Voigt et al., 2017*), substituting the Renilla sequence with different inserts (*Figure 1A and B*). We designed a proline-rich reporter (PPG) containing a COL1A1 subsequence rich in proline-proline-glycine (P/P/G) repeats, which are known to induce ribosome stalling and depend on eIF5A activity (*Barba-Aliaga et al., 2021*; *Gutierrez et al., 2013*; *Schuller et al., 2017*; *Pelechano and Alepuz, 2017*). As controls, we generated two additional reporters: one in which the proline codons in the COL1A1 subsequence were replaced with alanine (AAG), a neutral amino acid encoded by fast-translating codons (*Gobet et al., 2020*), to minimize stalling and possible eIF5A dependence; and another with no insert sequence (no-insert) as a control for SunTag protein synthesis. The original SunTag-Renilla-MS2 reporter (Renilla) was also included in our analysis (*Figure 1B*). Together, these four constructs allowed us to systematically compare the translation dynamics of mRNAs with varying coding sequences and to evaluate the impact of elongation perturbation on translation kinetics.

All reporters were stably integrated into HeLa cells under a doxycycline-inducible promoter (*Weidenfeld et al., 2009*). The cell line also stably expresses scFv-GFP antibodies against the SunTag together with NLS-stdMCP-stdHalo-RH1 recognizing the MS2 loop (MCP) while binding to the actin cortex (RH1), stabilizing the mRNA for long-term imaging (*Bhaskar et al., 2020*; *Figure 1C*).

Using a spinning-disk confocal microscope, we imaged translation over several minutes (*Figure 1— figure supplement 1A* and *Figure 1—video 1*), observing a highly dynamic and heterogeneous process characterized by active and inactive periods (*Figure 1D*). We quantified the duration of these periods empirically, by measuring the duration of silent (inactive) and translated (active) periods. In most experiments, more than 50% of mRNAs were translated (*Figure 1E*) with a mean trace duration exceeding 10 min (*Figure 1—figure supplement 1B*). We quantified the duration of translated and silent periods (*Figure 1F*), finding that active translation typically lasts around 10–15 min, interspersed with shorter silent periods of approximately 5–10 min across all reporters (*Figure 1G*). The slightly shorter translated periods observed for the no-insert reporter are likely due to its shorter length, which allows observation of isolated bursts more frequently.

We established an experimental method to measure the intensity of one mature protein, allowing us to estimate the number of translating ribosomes across translated periods. The measurement of mature proteins is challenging due to their rapid diffusion and low intensity against the high background of unbound scFv-sfGFPs. To overcome this, we used a SunTag-Renilla-RH1 construct to tether mature proteins to the actin cytoskeleton (*Voigt et al., 2017*). However, as these spots remained too dim at experimental laser powers, we developed an intensity-power calibration assay. This allowed us to measure spot intensity at high power and extrapolate the value for live-imaging conditions (14±2 a.u., Methods and *Figure 1—figure supplement 2A-C*). No-insert, AAG, and Renilla reporters showed on average less than 10 ribosomes per mRNA (*Figure 1—figure supplement 2D*), while the PPG reporter showed a slightly higher number (12 ribosomes per mRNA). Assuming a uniform ribosome distribution along the transcript during a translated period, these observations imply a low ribosome density, with 7–12% of the transcript length covered by ribosomes, on average (*Figure 1H*). However, given the observed bursting dynamics, the actual ribosome distribution may deviate from a uniform distribution, potentially resulting in higher local ribosome densities. Nevertheless, our data

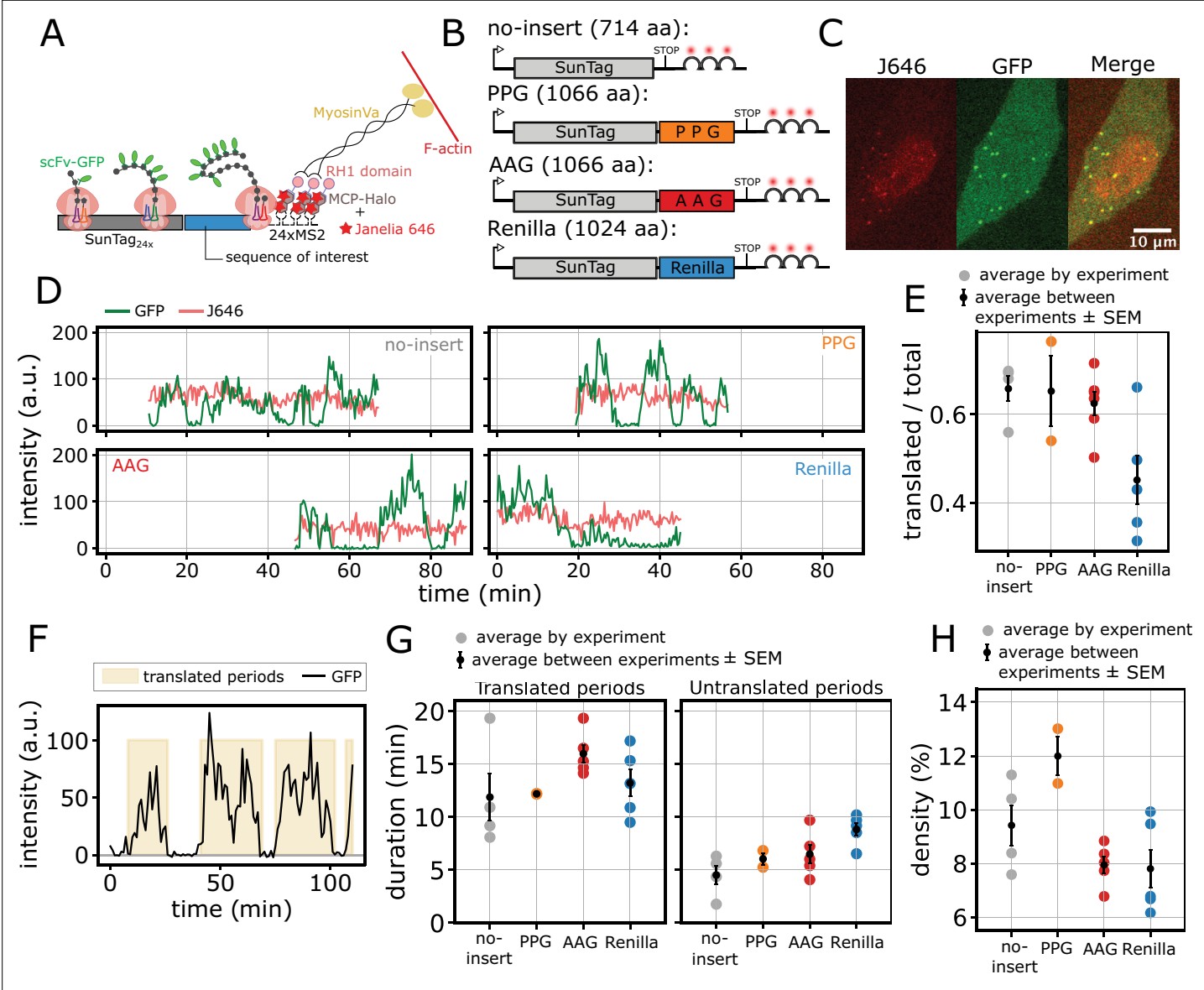

**Figure 1.** Translation exhibits bursts with low average ribosome density. (**A**) Schematic of the SunTag system used for single-molecule imaging of translation. (**B**) SunTag reporters differing in their coding sequence inserts: no insert, PPG (proline-rich), AAG (alanine-rich), and Renilla. (**C**) Representative live-cell image of a HeLa cell expressing the PPG reporter, showing JF646 (mRNA), GFP (nascent peptide), and merged signals. The image is part of a larger field of view, acquired with a 60 X objective in a spinning-disk confocal setup. Scale bar, 10 μm. (**D**) Representative long traces for each reporter, showing stable JF646 (mRNA, red) and fluctuating GFP (SunTag, green) intensities over time. (**E**) Average fraction of translated traces for each reporter. Colored dots represent averages from individual experiments; black dots with bars show the mean ± SEM across multiple experiments (*n* traces, *n* experiments): no-insert (669, 4), PPG (440, 2), AAG (1235, 6), Renilla (804, 5). (**F**) Example of the SunTag signal of a translated mRNA (no-insert), highlighting translated periods (shaded yellow). (**G**) Average duration of translated periods (left) and untranslated periods (right) for traces longer than 20 min. Colored dots represent averages from individual experiments; black dots with bars show the mean ± SEM across multiple experiments (*n* traces, *n* experiments): no-insert (41, 4), PPG (45, 2), AAG (90, 5), Renilla (62, 5). (**H**) Estimated average ribosome density (percentage of transcript covered by ribosomes) for traces in (**G**). Significance tests performed with the Mann-Whitney U test.

The online version of this article includes the following video and figure supplement(s) for figure 1:

**Figure supplement 1.** SunTag traces in control conditions.

**Figure supplement 2.** Experimental measurement of mature protein intensities to estimate number of translating ribosomes.

**Figure 1—video 1.** Time-lapse imaging of HeLa cells expressing the AAG reporter.

https://elifesciences.org/articles/107160/figures#fig1video1

show that, independently of the reporter sequence, bursting dynamics and ribosome densities were very similar, although this does not inform on the relative contributions of initiation and elongation to these observations. Finally, the observation of bursting behavior is in agreement with what was measured by *Livingston et al., 2023* for other SunTag reporters.

## TASEP-based inference of translation dynamics from single run-off traces

To gain more information on the translation dynamics, we performed harringtonine (HT) run-off assays, under the same imaging conditions (*Figure 2—figure supplement 1A*, *Figure 2—video 1; Figure 2—video 2*). HT blocks the first step of elongation after subunit joining, while allowing translating ribosomes to elongate and terminate (*Fresno et al., 1977*). The time required for the SunTag signal to disappear approximates the total elongation time but depends on the position of the last (most upstream) elongating ribosome when HT blocks initiation. Because the distribution of ribosomes on each transcript is unknown, it is common practice to average the signal from multiple traces to estimate the total elongation time (hence, the average ribosome speed) (*Yan et al., 2016*; *Wang et al., 2016*; *Mateju et al., 2020*; *Aguilera et al., 2019*). Although this pseudo-bulk approach has been successful in estimating average elongation times, it has the limitation of neglecting trace-to-trace heterogeneity.

Here we used a novel approach, allowing us to infer the number of translating ribosomes over time at the single-mRNA level. In particular, we used a Hidden Markov Model (HMM), where the hidden states represent the number of ribosomes translating the SunTag reporter during the run-off (*Figure 2A* and Materials and methods). The transition probabilities between hidden states were derived from an approximation of the Totally Asymmetric Exclusion Process (TASEP), assuming low ribosome density and homogeneous elongation rates. These probabilities define the likelihood of observing one or more termination events per imaging time step. This approach allowed us to relate the observed run-off times to the underlying initiation (α) and elongation rates ($\lambda$) (*Figure 2A*), as well as to decode single traces by inferring the number of translating ribosomes over time (*Figure 2B*). We assumed that HT takes 60 s to diffuse into the cells and effectively block initiation (*Ingolia et al., 2009*; *Aguilera et al., 2019*).

In our model, we accounted for the finite size of the SunTag by calculating correction factors for the intensity mean and variance ($\gamma(t)$ and $\nu(t)$, Methods). In particular, $\gamma(t)$ generalizes the time-independent correction factor previously derived by others (*Aguilera et al., 2019*) to the run-off dynamics (*Figure 2C* and Materials and methods). We first tested these approximations by comparing them with numerical simulations of the homogeneous $\ell$-TASEP (Materials and methods). We used different initiation and elongation rates, corresponding to a broad range of ribosome densities. These simulations confirmed the validity of our analytical approximations at low ribosome densities (*Figure 2C, D*, *Figure 2—figure supplement 2A*).

We then tested the performance of the model on simulated SunTag traces, with different values of the kinetic parameters and ribosome density ($\rho = 0.01 - 0.5$, i.e. average fraction of occupied transcript; *Figure 2E*). Instead of fixing the mature protein intensity $i_{MP}$, we used the model to infer its value, alongside α and $\lambda$. This parameter determines the extent of the intensity drop due to a termination event. The kinetic parameters and the mature protein intensity are accurately inferred at low density ($\rho \lesssim 10\%$) for different noise levels (*Figure 2E*), despite some correlation between α and $i_{MP}$ (*Figure 2—figure supplement 2B*). The correlation is more pronounced at higher density, where $i_{MP}$ is underestimated and α is overestimated. Importantly, the average ribosome density is overestimated at high densities – we are neglecting ribosome interference that can reduce the effective initiation rate – while it is accurately inferred at low densities (*Figure 2—figure supplement 2C*).

## Low ribosome density arises from coordinated translation initiation and elongation

After testing our model on simulated traces, we used it to analyze the HT run-off assays. To avoid correlations between parameters, we fixed the value of $i_{MP}$ to the experimentally measured value (14 ± 2 a.u.). We modeled intensity noise as lognormal and estimated its magnitude from cycloheximide-treated traces (Methods and *Figure 3—figure supplement 1*). The model allowed us to estimate the number of ribosomes translating the mRNA for each translation site, and thus the total run-off time

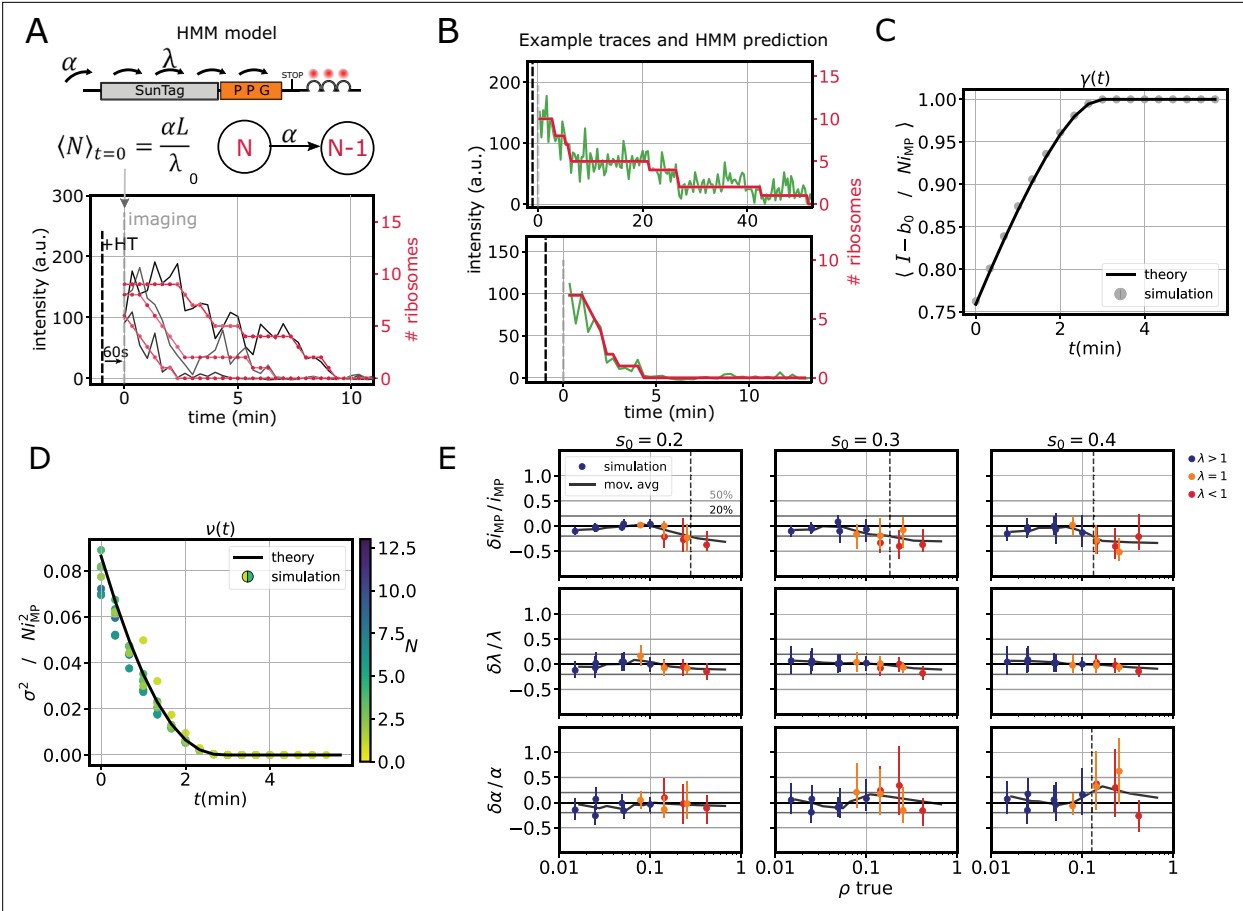

**Figure 2.** TASEP-based inference of translation dynamics from single run-off traces. (**A**) Schematic of the Hidden Markov Model (HMM) used to analyze run-off experiments. $\alpha$ = initiation rate, $\lambda$ = average elongation rate, $L$ = reporter length, $\langle N \rangle_{t=0}$ = average number of ribosomes at $t = 0$ (60 s after HT addition). (**B**) Representative slow (top) and fast (bottom) decaying traces from one run-off experiment with the PPG reporter; intensity in green and predicted number of ribosomes in red. (**C**) Comparison between the analytical correction factor $\gamma$ (black line, Materials and methods, *Equation 24*) and the results obtained by numerical simulations of run-off traces (gray dots). The latter represent $(I(N, t) - b_0)/Ni_{MP}$ averaged over 2000 simulated run-off traces, with $I(N, t)$ simulated intensity, $b_0$ offset, $N$ number of ribosomes, and $i_{MP}$ mature protein intensity. Kinetic parameters used in simulations: initiation $\alpha = 1/60 \, s^{-1}$, elongation $\lambda = 3.0 \, aa/s$. (**D**) Comparison between the analytical correction factor $\nu$ (black line, Materials and methods, *Equation 26*) and the results obtained by numerical simulations of run-off traces (colored dots). The latter represents the normalized intensity variance $\sigma^2(N, t)/Ni_{MP}^2$ for different values of $N$, same simulated traces as in **C**. (**E**) Relative error on model parameters ($\alpha$, $\lambda$, $i_{MP}$) vs average ribosome density $\rho$, for different values of the noise parameter $s_0$ (Materials and methods *Equation 44*), for simulated run-off traces. Each data point is obtained with a different combination of $\lambda \in \{0.5, 1, 3, 5\} \, aa/s$ and $\alpha \in \{1/120, 1/60, 1/30\} \, s^{-1}$. Blue dots represent the relative error on each parameter, the black curve is a moving average, and the dashed line indicates the density $\rho$ at which the relative error goes beyond 0.2. Simulation parameters in (**C**) – (**E**): $i_{MP} = 10$ a.u., reporter length $L = 1066 \, aa$, ribosome size $\ell = 10 \, aa$.

The online version of this article includes the following video and figure supplement(s) for figure 2:

**Figure supplement 1.** SunTag run-off traces in control conditions after HT addition.

**Figure supplement 2.** TASEP-based inference accurately estimates ribosome density on simulated data despite correlations between α and *i*MP.

**Figure 2—video 1.** Time-lapse imaging of HeLa cells expressing the PPG reporter after harringtonine treatment.
https://elifesciences.org/articles/107160/figures#fig2video1

**Figure 2—video 2.** Time-lapse imaging of HeLa cells expressing the AAG reporter after harringtonine treatment.
https://elifesciences.org/articles/107160/figures#fig2video2

for each trace (i.e. the time at which the number of translating ribosomes reaches zero; *Figure 3A*, violin plot). These single-trace estimates revealed substantial variability in run-off times, which likely reflects elongation heterogeneity and stalling events beyond what can be explained by differences in initial ribosome distribution.

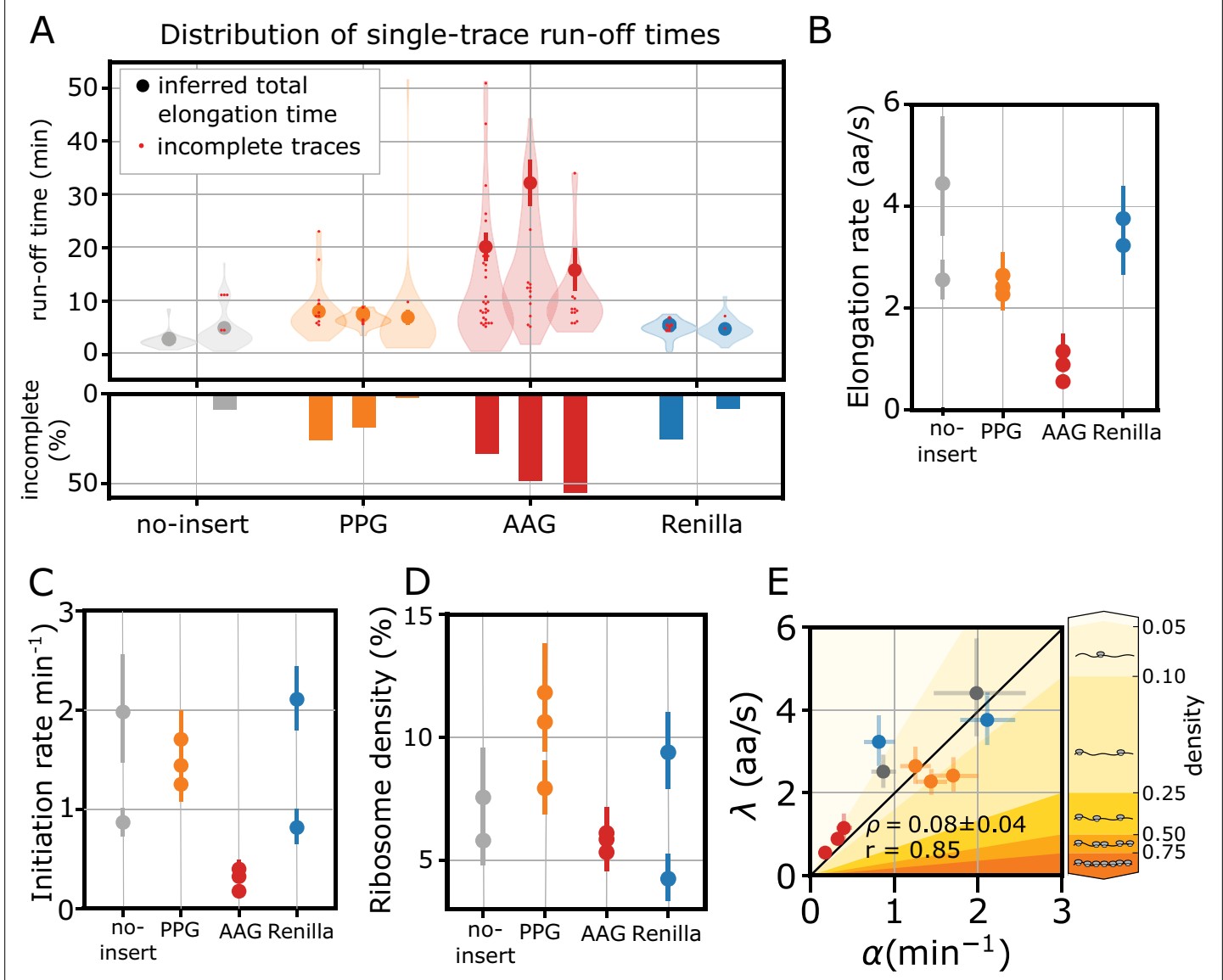

**Figure 3.** Low ribosome density arises from coordinated translation initiation and elongation. In all panels, the error bars indicate 95% confidence intervals. (**A**) Run-off time distribution (top) and percentage of incomplete run-offs (bottom) for each reporter. For incomplete runoffs, we include the total duration of the trace, as it represents a lower bound to the total run-off time (red dots). (**B**) Inferred elongation rates ($\lambda$). (**C**) Inferred initiation rates ($\alpha$). (**D**) Inferred average ribosome density. (**E**) Correlation between elongation and initiation rates for each experiment. The shaded background indicates approximate ribosome density regimes, ranging from low density (light yellow) to high density (dark yellow), as shown by the adjacent color bar. $\rho$ indicates the average density ± standard deviation, $r$ is the Pearson correlation coefficient.

The online version of this article includes the following figure supplement(s) for figure 3:

**Figure supplement 1.** Global measurement noise estimation from CHX traces.

**Figure supplement 2.** HMM approach predicts lower elongation speed than simple linear regression and suggests the presence of bursting and stalling.

However, for some traces, the mRNA signal was lost before run-off completion (*Figure 3A*, histogram). This occurred rarely for the no-insert reporter, while for the Renilla and PPG reporters, we observed about 10–20% of incomplete traces. Surprisingly, the AAG reporter showed up to 50% of incomplete traces per experiment and longer run-off times, suggesting considerable ribosome stalling (*Figure 3A*). The elongation rates inferred by our model for the no-insert, PPG, and Renilla reporters ranged from 2 to 4.5 aa/s, consistent with previous reports (2–6 aa/s; *Yan et al., 2016*; *Mateju et al., 2020*; *Livingston et al., 2023*; *Madern et al., 2025*; *Figure 3B*). As we expected from

the run-off times distribution, the AAG reporter was markedly slower, with elongation rates close to 1 aa/s (*Figure 3B*).

It is common practice (*Yan et al., 2016*; *Wang et al., 2016*; *Mateju et al., 2020*; *Aguilera et al., 2019*) to infer the total elongation time by fitting a linear decay to the average run-off intensity, usually discarding the slower decay resulting from heterogeneity in elongation (*Wang et al., 2016*). The intercept between the fast linear decay and the $x$-axis gives an estimate of the total average elongation time (*Figure 3—figure supplement 2A*). Our single-trace inference method tends to estimate lower elongation speeds with respect to what was obtained with the population-averaged linear fit (*Figure 3—figure supplement 2A, B*). This is likely because our model accounts for the initial ribosome distribution and is more sensitive to trace-to-trace translation heterogeneity. In addition, the HMM allowed us to decode the single-mRNA traces and estimate the time between consecutive termination events, which significantly deviated from an exponential distribution (*Figure 3—figure supplement 2C*). This can be partially explained by bursting; however, since initiation is inhibited upon HT treatment, waiting times between termination events that are much longer than the average elongation time most likely reflect persistent ribosome stalling.

The inferred initiation rates corresponded to ~1–2 ribosomes per minute (*Figure 3C*), close to what was measured in similar systems (1–5 ribosomes per min) (*Boersma et al., 2019*; *Barrington et al., 2023*). Surprisingly, the AAG reporter showed markedly lower initiation rates (*Figure 3C*). From the elongation and initiation rates, we calculated the average ribosome density under the low-density assumption (*Figure 3D*, Methods *Equation 14*). Despite the variability in elongation rates, we obtained low densities for all reporters, remarkably similar to our previous estimates (*Figure 1H*). Furthermore, we observed a significant correlation between initiation and elongation rates (Pearson correlation coefficient $r = 0.85$, average density $\rho = 0.08$; *Figure 3E*). This suggests a tight feedback between initiation and elongation, where changes in one rate are compensated by changes in the other to maintain a low ribosome density.

## eIF5A perturbations minimally affect ribosome density and translational bursting

Having established a correlation between initiation and elongation rates across reporters, we next aimed to investigate the impact of perturbing translation elongation on initiation and ribosome density. We chose to target eIF5A, a protein known to play a crucial role in ribosome elongation (*Schuller et al., 2017*; *Pelechano and Alepuz, 2017*; *Manjunath et al., 2019*). eIF5A activity was modulated using two distinct approaches. First, HeLa cells were incubated for 24 hr with 1 µM and 10 µM GC7. GC7 is a spermidine analog that inhibits Deoxyhypusine synthase (DHPS), the enzyme responsible for hypusination of eIF5A, thereby reducing the levels of active, hypusinated eIF5A (h-eIF5A) (*Barba-Aliaga et al., 2021*; *Giraud et al., 2020*; *Schultz et al., 2018*; *Coni et al., 2020*; *Szepesi et al., 2023*; *Matsumoto et al., 2023*). Upon treatment, h-eIF5A levels were reduced by ~ two- and ~ threefold, respectively, while total eIF5A levels remained unchanged (*Figure 4A*). To minimize potential side effects of GC7, we selected the 1 µM concentration throughout our experiments. As a second approach, we generated a CRISPR/Cas9 eIF5A knockout (KO) cell line derived from the PPG-expressing HeLa cells. This KO cell line exhibited a complete loss of both total and hypusinated eIF5A (*Figure 4B*).

Upon doxycycline (DOX) induction, both GC7-treated cells and eIF5A KO cells showed expression of the SunTag reporters and exhibited active translation (*Figure 4C-E*, *Figure 4—figure supplement 1A*, *Figure 4—video 1*; *Figure 4—video 2*), with no significant difference in the fraction of translated mRNA (*Figure 4F*) and translated trace duration (*Figure 4—figure supplement 1B*). As for control conditions, the translation signal showed bursting dynamics (*Figure 4E*), with duration of translated and untranslated periods similar to the control (*Figure 4G and H*). Ribosome density was remarkably stable between control and perturbed conditions, with no statistically significant differences (*Figure 4I*). In summary, these findings indicate that decreasing eIF5A activity or level has a minimal effect on translational bursting and ribosome density in our system.

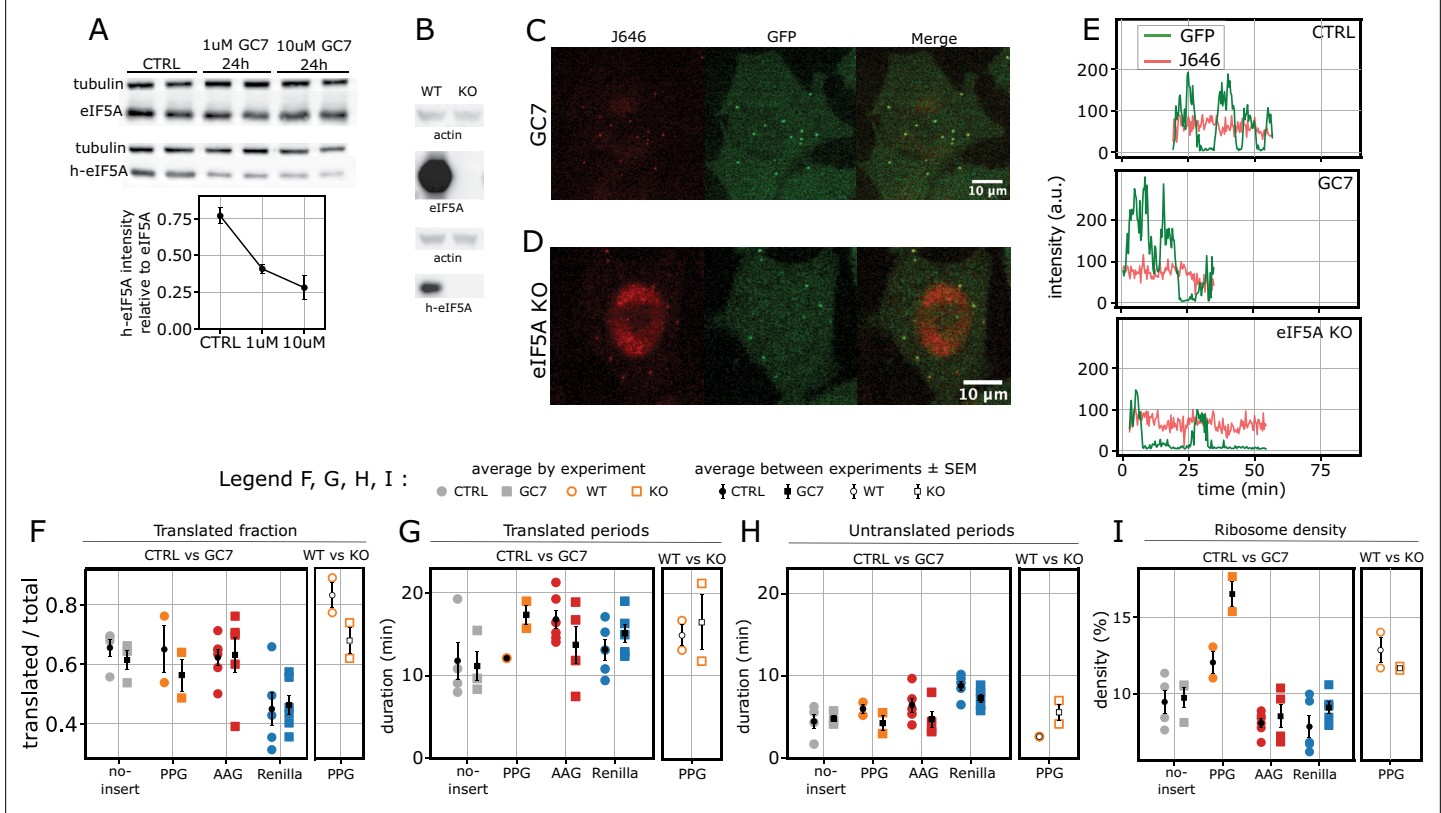

**Figure 4.** eIF5A perturbations minimally affect ribosome density and translational bursting. (**A**) Top: western blot of h-eIF5A levels in control cells and cells treated with 1 μM and 10 μM GC7 for 24 hr, with 1 mM AG. Bottom: quantification of h-eIF5A signal relative to total eIF5A. Error bars represent standard deviation between replicates. (**B**) Total eIF5A and h-eIF5A expression in CRISPR/Cas9 knock-out (KO) clones compared to wild-type (WT) HeLa cells. (**C**) Representative live-cell images of a HeLa cell expressing the PPG reporter after 24 hr treatment with 1 μM GC7 and 1 mM AG, showing JF646 (mRNA, red), GFP (SunTag, green), and merged signals. The image is part of a larger field of view, acquired with a 60 X objective in a spinning-disk confocal set-up. Scale bar, 10 μm. (**D**) Same as (**C**) for eIF5A KO cells. (**E**) Representative traces of the PPG reporter under control conditions, GC7 treatment, and eIF5A KO, showing GFP (SunTag, green) and JF646 (mRNA, red) intensities over time. (**F**) Fraction of translated mRNAs (*n* traces, *n* experiments): no-insert CTRL (669, 4), GC7 (511, 3); PPG CTRL (440, 2), GC7 (392, 2); AAG CTRL (1235, 6), GC7 (900, 5); Renilla CTRL (804, 5), GC7 (1476, 6); PPG WT (396, 2), KO (343, 2). (**G**) Average duration of translated periods and (**H**) untranslated periods for traces longer than 20 min. (**I**) Average ribosome density during the translated periods analyzed in (**G**). (**G – I**) (*n* traces > 20 min, *n* experiments): no-insert CTRL (41, 4), GC7 (20, 3); PPG CTRL (45, 2), GC7 (28, 2); AAG CTRL (90, 5), GC7 (26, 4); Renilla CTRL (62, 5), GC7 (112, 5); PPG WT (33, 2), KO (26, 2). Colored dots represent averages from individual experiments, black dots with bars show the mean ± SEM across multiple experiments. Mann-Whitney U significance test.

The online version of this article includes the following video and figure supplement(s) for figure 4:

**Figure 4—video 1.** Time-lapse imaging of HeLa cells expressing the PPG reporter after 24 hr GC7 treatment.

https://elifesciences.org/articles/107160/figures#fig4video1

**Figure 4—video 2.** Time-lapse imaging of EIF5A KO HeLa cells expressing the PPG reporter.

https://elifesciences.org/articles/107160/figures#fig4video2

**Figure supplement 1.** SunTag traces in perturbed conditions.

## eIF5A-driven elongation changes induce initiation regulation to preserve ribosome density

To understand how elongation and initiation contribute to maintaining low ribosome density upon eIF5A perturbations, we performed HT run-off experiments on GC7-treated and eIF5A KO cells, alongside control conditions (*Figure 5A*, *Figure 5—figure supplement 1A*; *Figure 5—videos 1–3*).

Using our TASEP-based model, we evaluated the intensity of a single mature protein by jointly fitting control and perturbed conditions with a shared $i_{MP}$ parameter, while still allowing condition-specific initiation and elongation rates (*Figure 5—figure supplement 2A*). Strikingly, the inferred $i_{MP}$ values closely matched the independently measured intensity of (14±2) a.u. (*Figure 5—figure supplement 2B*). This close quantitative agreement, combined with the absence of any significant

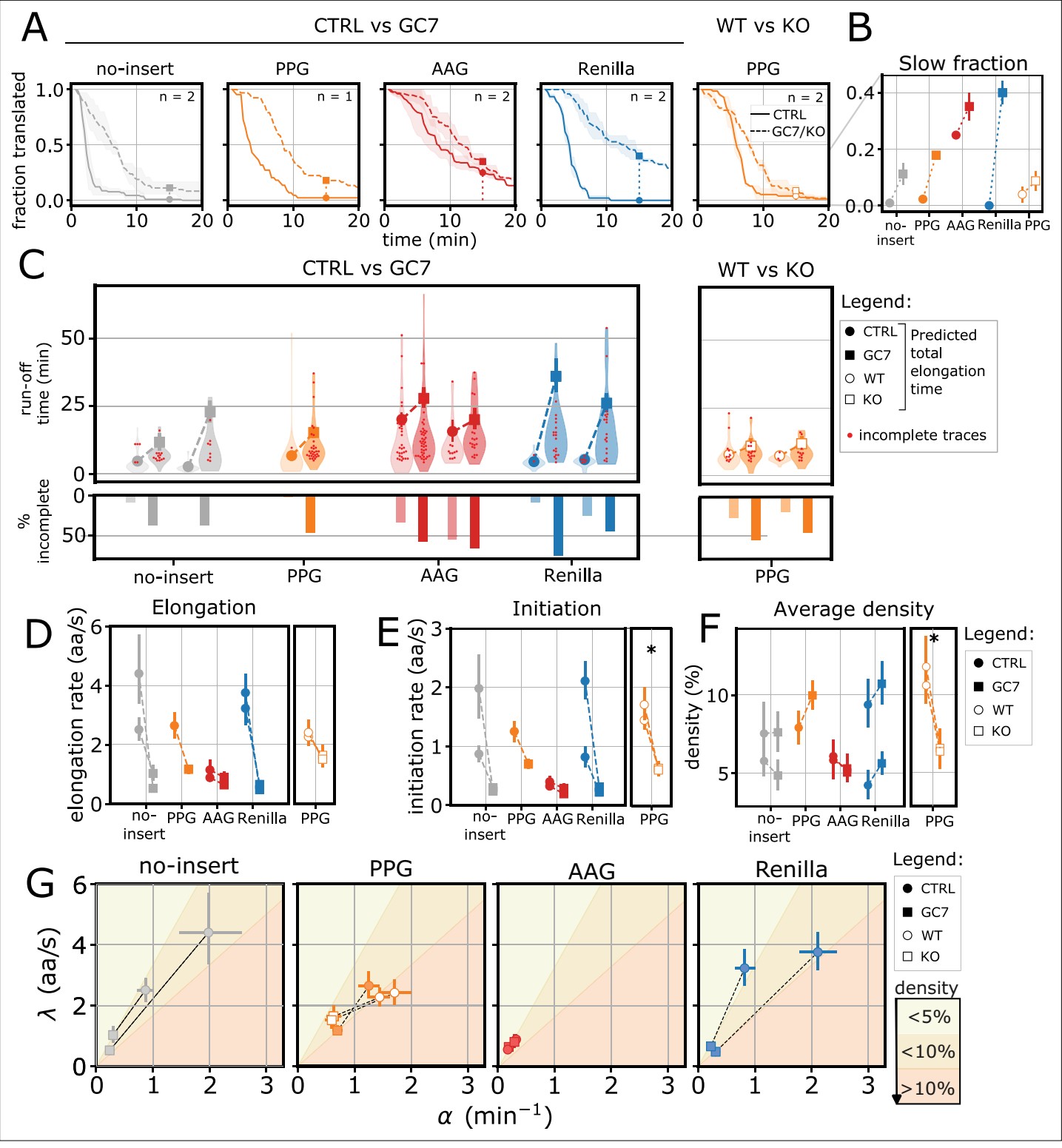

**Figure 5.** eIF5A-driven elongation changes induce initiation regulation to preserve ribosome density. (**A**) Fraction of translated mRNAs over time in control (solid lines) and perturbed conditions (GC7 treatment and eIF5A KO, dashed lines) for each reporter. Thick lines represent the mean across experiments, shaded areas represent standard deviation between experiments. Full circles and squares represent the fraction of mRNAs still translated 15 min after HT addition (control and perturbed conditions, respectively). (*n* is the number of experiments) (**B**) Fraction of translated traces showing slow run-off (still translated after 15 min of HT treatment). (**C**) Run-off time distribution (top) and percentage of incomplete run-offs (bottom) for each reporter in control conditions (circles) and perturbed conditions (GC7 treatment and eIF5A KO, squares). For incomplete runoffs, we include the total

*Figure 5 continued on next page*

*Figure 5 continued*

duration of the trace, as it represents a lower bound to the total run-off time (red dots). Number of translated traces (control, perturbed): no-insert (58, 35), (22, 24); PPG (45, 67); AAG (93, 80), (20, 32); Renilla (24, 24), (24, 40); PPG WT/KO (39, 28), (16, 25). (**D**) Inferred elongation rates. (**E**) Inferred initiation rates. (**F**) Inferred average ribosome density. (**G**) Representation of changes in translation kinetic parameters upon eIF5A perturbation in the initiation-elongation ($\alpha - \lambda$) plane. The shaded background indicates approximate ribosome density regimes, ranging from low density (light yellow) to high density (dark yellow), as shown by the adjacent color bar. Significance tests performed with the Mann-Whitney U test.

The online version of this article includes the following video and figure supplement(s) for figure 5:

**Figure 5—video 1.** Time-lapse imaging of HeLa cells expressing the PPG reporter after 24 hr GC7 treatment and harringtonine addition.
https://elifesciences.org/articles/107160/figures#fig5video1

**Figure 5—video 2.** Time-lapse imaging of HeLa cells expressing the AAG reporter after 24 hr GC7 treatment and harringtonine addition.
https://elifesciences.org/articles/107160/figures#fig5video2

**Figure 5—video 3.** Time-lapse imaging of EIF5A KO HeLa cells expressing the PPG reporter after harringtonine addition.
https://elifesciences.org/articles/107160/figures#fig5video3

**Figure supplement 1.** SunTag run-off traces in perturbed conditions after HT addition.

**Figure supplement 2.** Inferred mature protein intensities are in good agreement with the experimental measurement.

correlation between α and $i_{MP}$ (*Figure 5—figure supplement 2C*), indicates that the model recovers this parameter robustly rather than compensating for other rates. Given this strong internal consistency between measurement and inference, we fixed $i_{MP}$ to its experimentally determined value for all subsequent analyses.

After 24 hr of GC7 treatment, we observed an increase in the average run-off time across all reporters (*Figure 5A*). The subset of traces showing slow run-off (≥15 min) – < 4% in control conditions, with the exception of the AAG – increased significantly upon GC7 treatment (11% for no-insert, 18% for PPG, and 40% for Renilla; *Figure 5B*). The average total elongation time inferred with our model also increased for all reporters (*Figure 5C*). The extent of the increase varied across experiments for no-insert and Renilla: two- and eightfold changes for no-insert and eight- and fivefold changes for Renilla. More consistent, although milder increases were observed for PPG, both upon GC7 treatment (twofold change) and eIF5A KO (1.5-fold). The already slow AAG reporter was only mildly affected (1.4-fold change). Our HMM inference results confirmed a general decrease in elongation rates in both GC7-treated and eIF5A KO cells (*Figure 5D*). Importantly, we also observed a concurrent decrease in initiation rates under both perturbation conditions.

In GC7-treated cells, both elongation and initiation rates decreased proportionally (*Figure 5D and E*), resulting in a nearly constant ribosome density (*Figure 5F and G*). This is consistent with our previous observations (*Figure 4I*), which showed that average ribosome density was largely unaffected by the GC7 treatment. However, the eIF5A KO cells revealed a different outcome. While elongation rates also decreased in the KO, the magnitude of the decrease was less pronounced compared to GC7 treatment (*Figure 5D*). In particular, the fraction of traces showing a slow run-off (*Figure 5A*, right tail, and *Figure 5B*) was reduced in eIF5A KO with respect to GC7 treatment (18% in GC7 treatment and 9% in eIF5A KO). At the same time, initiation was strongly decreased (*Figure 5E*). As a result, we observed a significant reduction in average ribosome density in the eIF5A KO cells (*Figure 5F and G*, 30–40% decrease), which we did not observe upon GC7 treatment. This can be explained by either faster elongation or more efficient stalling resolution via RQC mechanisms (*Latallo et al., 2023*; *Goldman et al., 2021*). Overall, we observed that initiation is dynamically adjusted to elongation upon eIF5A perturbation, most likely to avoid ribosome crowding.

## Discussion

By integrating single-mRNA imaging with single-trace statistical modeling, our study highlights key principles of translational regulation. First, we demonstrate that translation maintains remarkably low ribosome density (≤12% of the CDS occupied by ribosomes) across diverse coding sequences by coordinating initiation and elongation rates. Second, when elongation is perturbed by modulating eIF5A activity, initiation is proportionally adjusted to preserve low-density translation. Third, complete eIF5A knockout (KO) alters this coordination, suggesting differences in the sensing of ribosome stalling or collisions in the absence of eIF5A.

Our results support a model in which translation under homeostatic conditions occurs at low ribosome density, with initiation serving as the rate-limiting step (*Riba et al., 2019*; *Boersma et al., 2019*; *Livingston et al., 2023*). Indeed, using the SunTag system to visualize translation of single mRNAs, we consistently observed low density across all reporter mRNAs, regardless of coding sequence variations. This conserved feature may reflect an evolutionary strategy aimed at avoiding overuse of resources, as the same protein synthesis rate can be attained by engaging fewer ribosomes at the same time (*Erdmann-Pham et al., 2020*). Since the pool of free ribosomes may limit initiation (*Shah et al., 2013*), maximizing this pool may be beneficial for the cell, since it allows more rapid and flexible adaptation of initiation rates in response to changing demands. At the same time, by minimizing the frequency of transient, stochastic collisions that arise from high ribosome density, the cell enhances its ability to detect and respond to problematic translation events. Persistent ribosome collisions can serve as clear signals of aberrant translation and trigger mRNA degradation, in agreement with the observation that high ribosome occupancy can destabilize mRNAs (*Presnyak et al., 2015*; *Bazzini et al., 2016*; *Narula et al., 2019*; *Wu et al., 2019*; *Bae and Coller, 2022*; *Bicknell et al., 2024*).

The low ribosome density across reporters stems from the dynamic interplay between elongation and initiation rates. Even though our mRNAs shared an identical 5' UTR, we observed significant and coupled variations in elongation and initiation across different coding sequences. It is important to note that this correlation is not a consequence of our model or inference, as the results on simulated data showed that we could infer a wide range of initiation and elongation rates, corresponding to very different densities. The strong correlation observed in real data suggests a feedback mechanism in which initiation is modulated by the prevailing elongation status, thereby preventing ribosome crowding and maintaining low density. The dynamic interplay between elongation and initiation has also recently been suggested in yeast (*Lyons et al., 2023*), in the case of mRNA reporters encoding synonymous codons. Similar conclusions have been reached in flies and humans for synonymous reporters, pointing to feedback mechanisms on translation initiation (*Barrington et al., 2023*). The main hypothesis underlying these observations in mammals is that suboptimal elongation leads to ribosome collisions, which are in turn sensed via molecular pathways that inhibit initiation.

The first response to ribosome collisions is the recruitment of the GIGYF2/4EHP complex, a cis-acting translational repressor that inhibits eIF4E binding and thus blocks further initiation (*Hickey et al., 2020*; *Juszkiewicz et al., 2020*). The GIGYF2/4EHP complex can rapidly repress initiation upon the detection of collisions (whereas more persistent collisions initiate RQC, e.g. ZNF598-mediated degradation; *Wu et al., 2020*). This molecular feedback on initiation, being transcript-specific, could explain our observations, contrary to a more general response to collisions such as the Integrated Stress Response (ISR) and eIF2 phosphorylation, which would globally attenuate initiation (*Wu et al., 2020*). It is important to note that our reporters differ in codon content only far downstream of the translation start site (about 600 aa), which allows us to exclude the hypothesis of ribosomes queuing up to the initiation site and physically blocking initiation. Indeed, this would result in a high ribosome density, in contrast to our observations.

If collisions are long-lived, alternative molecular pathways, for example involving ZNF598, can induce RQC and, eventually, mRNA degradation. Also, beyond ribosome collisions, other mechanisms were shown to detect slow-elongating ribosomes in yeast, inducing ribosome recycling and mRNA degradation (*Buschauer et al., 2020*; *Li et al., 2022*) – the evidence of similar pathways in mammals is still limited (*Absmeier et al., 2022*). Although our reporters showed a broad range of elongation rates (1–4 aa/s), there was no evidence of translation-dependent mRNA degradation, as we observed relatively consistent fractions of translated transcripts and trace duration distributions across reporters.

eIF5A plays a critical role in promoting elongation, especially in proline-rich sequences such as *COL1A1*. Thus, we next examined how perturbing eIF5A influences elongation, initiation, and ribosome density. Remarkably, significant reductions of active eIF5A – via GC7 treatment or genetic knockout – maintained largely consistent bursting dynamics and average ribosome densities compared to controls. However, run-off assays revealed different underlying mechanisms. GC7 treatment led to a coordinated decrease in both elongation and initiation rates, maintaining low ribosome density. Surprisingly, we observed stronger changes in run-off profiles for no-insert and Renilla compared to PPG, contrary to what was expected, as PPG contains known eIF5A-target motifs. It is possible that stronger stalling or collisions in PPG were masked by the RQC-mediated degradation of the SunTag reporter. Such degradation would lead to faster decay kinetics and could result in incorrect

interpretation of the apparent translation dynamics (*Latallo et al., 2023*). Nevertheless, our perturbation experiment suggests that moderately impaired elongation due to reduced eIF5A activity triggers a feedback loop that dynamically adjusts initiation downwards, maintaining low-density translation.

eIF5A KO appeared to affect translation differently than GC7 treatment, leading to a significant (30–40%) reduction in ribosome density due to a stronger decrease in initiation than elongation. In particular, eIF5A KO showed fewer traces showing slow run-off compared to GC7, resulting in a milder decrease in elongation rate relative to control conditions. Although we do not have a definitive explanation for this observation, we hypothesize that the strong reduction in total eIF5A levels could alter the kinetics of stalling recognition and ribosome degradation. In yeast, eIF5A is known to compete with Not5 (CNOT3 in human) (*Absmeier et al., 2022*) for binding to the ribosomal E site (*Buschauer et al., 2020*). When peptidyl transfer is slow, eIF5A can occupy the E site and resolve stalling by promoting peptide-bond formation. Conversely, when decoding is slow, both the A and E sites remain unoccupied, allowing Not5 to bind the E site and mark the ribosome for degradation. Recently, a similar mechanism was suggested in mammals (*Absmeier et al., 2022*). In this scenario, the absence of eIF5A could lead to more frequent or faster ribosome degradation, reducing the accumulation of stalled ribosomes and giving the appearance of a higher elongation rate. Ribosome recycling in normal conditions was recently estimated to take 22 min on average (*Madern et al., 2025*), but could be faster in the absence of eIF5A.

Interestingly, our attempt to optimize translation through proline-to-alanine substitutions in the COL1A1 PPG motif (AAG reporter) resulted in markedly slower elongation rates (~1 aa/s) and increased frequency of incomplete run-off traces (50%). This suggests that expressing non-native sequences, even with seemingly favorable substitutions and a comparable codon adaptation index, can introduce unanticipated challenges, such as impaired RNA secondary structure or co-translational folding constraints. Specifically, replacing prolines with alanines reduced elongation by 60%, likely due to disrupted co-translational folding. Given that prolines in collagen are critical for triple-helix formation (*Shoulders and Raines, 2009*), proline-to-alanine substitutions likely generate misfolded intermediates that stall ribosomes (*Barba-Aliaga et al., 2021*; *Komar et al., 2024*).

## Limitations

First, our observations derive from exogenous mRNA, which may not fully recapitulate endogenous collagen translation dynamics, particularly given the absence of ER-coupled processing and secretory pathway targeting. This difference may explain the limited GC7 effects on the SunTag reporter, as recent studies suggest a direct role for eIF5A in ER-coupled collagen translation and translocation (*Rossi et al., 2014*; *Mandal et al., 2016*; *Barba-Aliaga et al., 2021*). This limitation could be partially overcome by genetically encoding the SunTag sequence via CRISPR/Cas9 genome editing, as performed by *Dufourt et al., 2021* in *Drosophila* embryos. In addition, using an ER-specific SunTag reporter, as done in *Choi et al., 2026* by adding a trans-membrane domain, would better recapitulate ER-coupled translation.

Second, while decoding individual traces, our model infers shared (population-level) rates. Inferring transcript-specific parameters would be more informative, but it is highly challenging due to the uncertainty regarding the initial ribosome distribution on single transcripts. Pooling multiple transcripts together allows some assumptions on the initial distribution that are essential for the inference of average elongation and initiation-rate parameters, while revealing substantial mRNA-to-mRNA variability in the posterior decoding (e.g. *Figure 3—figure supplement 2C*).

The deviation from an exponential distribution in the waiting times between termination events could arise from ribosome stalling, bursting, or a combination of both. Each of these effects would tend to bias our inference toward underestimating the true initiation rate. Incorporating explicit models of stalling and bursty initiation in future work would help make the inference framework more robust. In addition, our current model infers a single effective elongation rate per reporter, which averages ribosome dynamics across both the SunTag region and the variable CDS. As a consequence, it may underestimate true elongation-rate differences between the sequences of interest. This limitation could be alleviated by introducing distinct elongation parameters for the SunTag and the CDS, and by sharing the SunTag elongation parameter across different reporters. Incorporating all these features (stalling, bursting, SunTag-specific elongation rate) into the TASEP simulations would also make the simulated data more realistic.

Last, run-off assays are inherently variable and can underestimate or overestimate elongation rates, especially with uncertainties in harringtonine pharmacokinetics (*Aguilera et al., 2019*) or RQC-mediated termination events (*Latallo et al., 2023*; *Goldman et al., 2021*).

## Outlook

An important contribution of our work is providing an initial framework for modeling translation kinetics at the single-trace level. Currently, this approach is tractable for run-off assays, where mRNAs eventually deplete their ribosomes, simplifying parameter estimation. Although challenging, extending the model beyond harringtonine assays would circumvent these limitations, while giving insights on initiation regulation, including bursting. This could be done by extending the model to steady-state translation and incorporating a two-state telegraph model to capture 'bursty' initiation. Furthermore, introducing constructs with different 5' UTRs, similar to the work of *Livingston et al., 2023*, would provide valuable insights into how bursting dynamics interact with elongation rates. In addition, the current model infers population level parameters while accounting for single-trace stochasticity, extending the model (hierarchically) to allow for transcript-specific rates or stalling could provide valuable insights.

Addressing the current limitations, including the influence of RQC on run-off kinetics and the non-exponential distribution of termination events, will further refine our understanding of translation dynamics and improve model accuracy. In parallel, integrating run-off and bursting traces within a unified framework and incorporating transcript-specific kinetic parameters should offer a more complete picture of how cells regulate translation at the single-mRNA level. The latter could potentially be achieved by replacing the HMM with a particle-filter approach, to infer single-transcript kinetics (*Wills and Schön, 2023*). Finally, exploring the molecular mechanisms underlying the observed coupling of initiation and elongation, as well as the functional significance of bursting dynamics and ribosome density regulation, will be critical for unraveling how cells maintain proteostasis under diverse physiological and stress conditions.

## Materials and methods

### Experimental design

#### Model system and cell culture

We used a cell line derived from the HeLa-11ht cell line generated by *Weidenfeld et al., 2009* which contains a site for Flp-RMCE (recombinase-mediated cassette exchange), allowing a single-copy genomic integration of a target gene, and constitutively expressing the reverse tetracycline-controlled transactivator (rtTA2-M2) for inducible expression. The parental cell line (HeLa-11ht) was authenticated by STR profiling (Eurofins Medigenomix Forensik GmbH) and confirmed to match the HeLa reference profile (DSMZ/Cellosaurus databases). All cell lines were tested negative for mycoplasma contamination. The cell line was further engineered to stably express GFP-tagged single-chain antibodies (scFv-GFP) against GCN4 and NLS-stdMCP-stdHalo-RH1 fusion protein (*Voigt et al., 2017*; courtesy of Jeffrey Chao Lab, FMI Basel). The MCP protein recognizes a MS2 stem-loop cassette in the 3' UTR of the reporter mRNA, while the RH1 domain associates with actin filaments through the interaction with the MyosinVa molecular motor allowing stable imaging of actin-anchored mRNAs (*Figure 1A*). In the following, we refer to this cell line as the parental cell line, from which we obtain monoclonal cell lines expressing the SunTag reporters by stable transfection and fluorescence-activated cell sorting (FACS).

HeLa cells are cultured in Dulbecco's Modified Eagle Medium (DMEM) containing 4.5 g/l glucose, Penicillin (100 U/ml), Streptomycin (100 µg/ml), L-Glutamine (4 mM) and 10% fetal bovine serum (FBS). Cells are maintained at 37 °C and 5% CO2. For transient and stable transfections, we use Fugene HD Transfection Reagent (Promega) with Opti-MEM reduced serum medium (Thermo Fisher Scientific) according to the manufacturer's instructions.

To perturb eIF5A activity, cells are treated with 1 µM N1-guanyl-1,7-diaminoheptane (GC7) for 24 hr before imaging. Because GC7 is known to be inactivated by the action of amine oxidases, which are abundant in serum, we add 1 mM aminoguanidine (AG) to inhibit amine oxidases, as is typically done in the literature (*Maier et al., 2010*). The control sample is treated with AG only.

## DNA constructs

All the constructs described in this project are obtained from the previously described SunTag-Renilla-MS2 plasmid (*Wilbertz et al., 2019*) (Addgene plasmid #119945, *Figure 1A*) via restriction-free cloning. The resulting plasmids contain SunTag (24 x GCN4), destabilized FKBP domain (*Banaszynski et al., 2006*) with a stop codon and a 24xMS2 stem-loop cassette. In addition, PPG contains a subsequence of Human collagen type I α1 (transcript COL1A1-201, from nucleotide 2619–3672, 1056 nt), while AAG contains the same subsequence where all proline amino acids (24% of the total amino acid content) are mutated to alanine, maintaining the same codon adaptation index (*Figure 1B*). The complete nucleotide sequences of the COL1A1 and mutated COL1A1 inserts are provided in *Supplementary file 1*.

SunTag-Renilla-RH1-MS2 is a plasmid containing the RH1 domain at the C-terminus of the Renilla open reading frame (*Voigt et al., 2017*), and was used in the calibration experiment to image the mature proteins.

## Cell line generation

$3 \cdot 10^5$ HeLa cells are seeded into a 6-well plate and the next day transfected with 2 μg of the FLPe recombinase plasmid (Addgene #20733) together with 2 μg of the plasmid carrying the SunTag reporter flanked by Flp-recombinase target sites. The next day, 5 μg/ml puromycin (Invivogen) is added to select for transfected cells. Two days later, the puromycin-containing medium is removed and cells are kept in growth medium containing 50 μM ganciclovir (Sigma-Aldrich) for 10 days to select for cells with successful RMCE. Single clones are isolated in a 96-well plate via FACS, based on cytoplasmic GFP intensity. Clones with moderate GFP background and responsive to doxycycline induction are selected for expansion.

## Transient transfection

For the calibration experiment, cells are transiently transfected with SunTag-Renilla-RH1-MS2 plasmid. Cells are seeded at $7.5 \cdot 10^3$ cells/well in a 8-well glass-bottom slide (170 μm, ibidi) 48 hr before imaging. After 24 hr, cells are transfected with Fugene transfection reagent according to the manufacturer's instructions with the SunTag plasmid but without the FLPe recombinase plasmid.

## Generation of eIF5A knock-out cell line

Starting from the cell line stably expressing PPG, we generate an eIF5A knock-out (KO) line using the single-guide RNA reported by *Manjunath et al., 2019* (eIF5A sgRNA #1 5' – AGAGGACCTTCGTCTC CCTG – 3') in an all-in-one Cas9-GFP plasmid. Single clones are selected with FACS sorting based on GFP intensity. Three clones are selected for further expansion, genotyping of the target site, and western blotting of eIF5A and h-eIF5A. The clone with the strongest reduction in total eIF5A is chosen for imaging.

## Western blot

Protein lysates were prepared by resuspending cell pellets in a hot SDS lysis buffer (2% SDS, 50 mM Tris-HCl, pH 7.5, 1 mM EDTA) and heating samples at 95 °C for 5 min. Samples were centrifuged for 5 min at maximum speed, and the supernatant was transferred to a new tube. Protein concentrations were measured using Pierce BCA Protein Assay Kit (Thermo Fisher Scientific) on a Tecan microplate reader. Protein lysates were mixed with 4 x LDS sample buffer (GenScript) and 100 mM DTT. Equal amounts of protein (7.5–15 μg per lane) were separated on 4–12% gradient SDS-PAGE gels (SurePAGE, Bis-Tris; GenScript) and transferred onto PVDF membranes using the iBlot 2 Dry Blotting System with PVDF transfer stacks (Invitrogen). Membranes were blocked in 5% milk in TBS-T (TBS with 0.1% Tween-20) for 1 hr at room temperature. Primary antibody incubation was performed overnight at 4 °C in 5% milk in TBS-T against: eIF5A (BD Biosciences, 611977, 1:20000, mouse), hypusinated eIF5A (mAbHpu24.B, Genentech, 1:3000, rabbit), α-tubulin (Cell Signaling Technology, 2144, 1:1000, rabbit) and β-actin (Cell Signaling Technology, 4967, 1:1000, rabbit). Anti-αtubulin was used as loading control for GC7 experiments, and anti-βactin for eIF5A KO experiments. After washing three times with TBS-T, membranes were incubated with HRP-conjugated secondary antibodies for 1 hr at room temperature: goat anti-mouse IgG-HRP (Cell Signaling Technology, 7076, 1:3000) for

**Table 1.** Technical specification of the live-cell imaging set-up.

| | |
|---|---|
| **Camera:** | **EMCCD Grayscale (ImagEMX2)**<br>**512x512 pixels, (16.0x16.5) μm/pixel**<br>**16-bit** |
| Objective: | U PLAN S APO 60x1.42NA |
| Laser lines: | 488 nm 640 nm |
| Emission filters: | sdc GFP BP 525/50 (Chroma ET525/50 m)<br>sdc Cy5 BP 700/75 (Chroma ET700/75 m) |

eIF5A, or goat anti-rabbit IgG-HRP (Cell Signaling Technology, 7074, 1:3000) for hypusinated eIF5A and loading controls. Chemiluminescent signals were detected using the Fusion FX imaging system (Vilber). Protein band intensities were quantified using ImageJ software, and target protein levels were normalized to the corresponding loading control.

## Live-cell imaging

Wild-type cells are seeded at a concentration of $7.5 \cdot 10^3$ cells/well in a 8-well glass-bottom slide (170 μm, ibidi) 48 hr before imaging. eIF5A knock-out cells are seeded at $15 \cdot 10^3$ to reach a similar confluency at the acquisition. For the experiments involving GC7 treatment, 24 hr before imaging the growth medium is replaced with fresh medium supplemented with 1 mM AG in the control wells and with 1 μM GC7 and 1 mM AG in the treated wells. For experiments involving the eIF5A knockout, the growth medium is replaced with fresh medium (without AG). Before imaging, the growth medium is replaced with fresh medium containing 1 μg/ml doxycycline (DOX) in order to induce the transcription of reporter mRNAs. After 40 min, the medium is further supplemented with Janelia Fluor 646 HaloTag ligand (JF646) (Promega), at 100 nM final concentration. After 20 min of incubation, the medium is removed and cells were washed in PBS. Cells are then kept in FluoroBrite DMEM (Thermo Fisher Scientific) supplemented with 10% FBS and 4 mM L-glutamine for 40 min before imaging and during the acquisition.

Cells are imaged using a spinning-disk confocal microscope (Visitron Spinning Disk CSU W1) equipped for live-cell imaging. Images are acquired using Visiview (Visitron) as single planes every 20 s and for 90 min, with 100ms exposure time, 60 X objective, and hardware autofocus (see *Table 1* for technical specifications). The focus is always set 0.8 μm away from the coverslip, yielding a focal plane close to the plasma membrane, ventral side of the cell. Because of the actin cortex, this area is usually rich in tethered reporter mRNAs. Cells are imaged sequentially in GFP and Cy5 channels with a single EMCCD camera.

In run-off assays, harringtonine (HT; Cayman Chemical) is pre-diluted in FluoroBrite and added to the cells (final concentration 3 μg/ml) mounted on the microscope stage. In practice, we add 53 μl of 20 μg/l working solution to 300 μl of FluoroBrite medium, pipetting up and down three times to ensure mixing. Image acquisition is started 30s – 60s after treatment (the delay between treatment and imaging is recorded for each acquisition). Time points are acquired every 20 s.

## Laser power measurement and flat-field correction

488 nm and 640 nm laser power is measured at the beginning of each experiment using a power meter and a 10 X Air objective (0.4 NA). During each acquisition – except in the calibration experiment – the percentage of laser power used is set to match 0.68 mW for the 488 nm laser and 2.5 mW for the 640 nm laser.

Every 2 months, both microscope flat field and dark current are measured. For the flat-field measurement, a uniformly fluorescent sample is prepared using ATTO 488 (AD 488–21, Atto-Tech) and ATTO 647 (AD 647–21, Atto-Tech), both in free acid form. Both dyes are dissolved in water at 1 mM concentration (ATTO 647 dissolves inefficiently, and the procedure can be improved by considering a different solvent) and then filtered with a 0.2 μm filter to remove aggregates. The solutions are added to 2% agarose gel to obtain a final concentration of 10 μM. The two fluorescent solutions are mixed 1:1 and a 40 μL drop is added to a microscopy slide and covered with a 22x22 mm coverslip, creating a thin, uniformly fluorescent, agarose layer. After drying, 20 xy positions are acquired in both

channels with the 60X oil objective. The final flat-field image is obtained by averaging the 20 images and applying a median filter with radius 5 pixels. For the dark field measurement, 100 time points are acquired without laser and without any specimen in both channels and averaged to obtain the dark current image.

## Mature protein intensity measurement

To measure the intensity of one mature protein (24 GFP molecules), we use a SunTag-Renilla-RH1-MS2 reporter with an RH1 domain at the end of the coding sequence for actin-tethering of mature proteins (*Voigt et al., 2017*). The cells are seeded and prepared for imaging as previously described, except that DOX incubation is shortened to 20 min to avoid overcrowding of mature proteins in the cytoplasm. Prior to imaging, cells are treated with 100 µg/ml of puromycin (Sigma-Aldrich) to disassemble translating ribosomes, then imaged at 100% power (4.63 mW measured with a 10 X Air objective) for 100 s at 10 Hz (*Figure 1—figure supplement 2A*). Spot detection, tracking, and intensity quantification (*Figure 1—figure supplement 2B*) are performed in the GFP channel only, since mature proteins do not colocalize with the mRNA signal. The intensity of each spot is averaged over the 100 frames.

To build the intensity-power calibration curve, cells are treated with cycloheximide (Sigma-Aldrich) at 200 µg/ml final concentration 5 min before imaging, to block the ribosomes and have constant GFP intensity per mRNA molecule. Cells are imaged at 20%, 40%, 60%, 80%, and 100% power, in rapid succession (10 frames per laser power at 0.5 Hz). This procedure revealed a linear relationship between the spot intensity and laser power (*Figure 1—figure supplement 2C*), allowing us to infer the mature protein intensity at the laser power used for live imaging, from the value measured at 100% laser power. The final estimate (14±2 a.u.) allows us to estimate the average number of ribosomes translating each report (*Figure 1—figure supplement 2D*).

## Quantification of intensity traces

### Image processing

The live-cell images acquired during experiments are corrected for flat field and dark current as follows

$$C = \frac{O - DC}{FF - DC} \langle FF - DC \rangle \tag{1}$$

where $C$ is the corrected image, $O$ is the original image, $DC$ is the dark current image, $FF$ is the flat field image and the average $\langle \ \rangle$ is calculated over all the pixels. Each experiment is corrected with the most recent dark current and flat-field images up to that date. Both the dark current and the flat-field measurements are very consistent across several months of imaging.

### Spot tracking and intensity quantification

The spot detection is performed in the far red channel with TrackMate 7 (Fiji) (*Ershov et al., 2022*). Nuclei were excluded from detection because of the very high background due to JF646 nuclear localization. The tracking is performed using the TrackMate simple Linear Assignment Problem (LAP) tracker, with a maximal linking distance of 2 µm, a maximal distance for gap-closing of 3 µm and a maximal frame interval between two linked spots of 2 frames. The minimal track length is set to 5 min.

Given a 2 µm × 2 µm region around an mRNA, the intensity of the spots in the far red and green channel is inferred using Bayesian inference. The spot intensity in each channel is approximated as a 2D Gaussian:

$$\mu_{2D}(x, y) = \frac{I}{2\pi w^2} \exp\left(-\frac{(x - x_0)^2}{2w^2} - \frac{(y - y_0)^2}{2w^2}\right) + g(x, y) \tag{2}$$

where $I$ is the total spot intensity, $w$ is the width of the spot, $x_0$ and $y_0$ are the coordinates of the spot center in micrometer units and $g(x, y)$ is a background given by

$$g(x, y) = c + b(y - y_0) + a(x - x_0), \tag{3}$$

with $a, b, c \in \mathbb{R}$. Hence, the background is modeled as a 2D plane with $a$ and $b$ representing the tilt in the $x$ and $y$ directions, respectively. This choice allowed us to model local changes in the cytoplasmic background, such as those happening when a spot is close to the cell membrane, and the GFP

background is higher inside the cell than outside. Both $a, b$, and $c$ are parameters of the model. Finally, the intensity of a pixel is modeled as

$$i(x, y) \sim \text{Normal}\left(\mu_{2D}(x, y), \sigma\right) \tag{4}$$

where $\sigma$ is the standard deviation and it is estimated as the pixel standard deviation in the cytoplasmic background, and its value is fixed during the inference.

The priors on $x_0$ and $y_0$ are centered on the position $x_r$, $y_r$ of the red spot, as given by TrackMate:

$$x_0 \sim \text{Normal}(x_r, 0.2)$$
$$y_0 \sim \text{Normal}(y_r, 0.2) \tag{5}$$

with a standard deviation of 0.2 μm approximately corresponding to the average distance between the mRNA spot and the translation spot. The prior on the spot size is very peaked for regularization purposes

$$w \sim \text{InvGamma}(\mu = 0.20, \sigma = 0.02). \tag{6}$$

The prior on the total intensity $I$ is an 'uninformative' prior

$$I \sim \text{HalfNormal}(\sigma = 1000). \tag{7}$$

The background $g(x, y)$ is a 2D plane, with

$$c \sim \text{Normal}(\mu_c, \sigma_c) \tag{8}$$

where $\mu_c$ and $\sigma_c$ are input parameters shared among all acquisitions, and they are estimated as the average and the standard deviation of the cytoplasm intensity across different cells and different experiments. Finally,

$$a, b \sim \text{Normal}(0, 50) \tag{9}$$

which we include to account for the background variation when a spot is close to the transition between the cell interior and exterior. For the calibration experiment described in Mature protein intensity measurement, all the steps (detection, tracking, and quantification) are done exclusively in the green channel, since mature proteins do not co-localize with the mRNA signal.

The inference is performed with the Hamiltonian Monte Carlo algorithm implemented in Stan (*Team, 2024*) through the PyStan Python interface.

## Preprocessing and data filtering

The estimated spot intensity is affected by tracking errors and diffusion of the mRNA in the $z$ direction of the focal plane. Cycloheximide (CHX) traces – acquired in the same imaging conditions as in Live-cell imaging – are used to estimate this noise and partially correct experimental traces. After regressing out the linear decay due to photobleaching, the standard deviation of the signal is calculated relative to the mean intensity of the trace; averaging over 236 traces yields a global estimate of the multiplicative noise (see *Figure 2—figure supplement 1* for examples of CHX traces and autocorrelation).

Next, all traces are smoothed with a low-pass filter (Butterworth filter), with a 1/120 Hz cut-off. At each time point, the relative difference between the raw and smoothed trace is calculated; if this relative difference is higher than twice the multiplicative noise calculated from cycloheximide traces, we replace it with the smoothed value. In practice, this corresponds to spikes or drops larger than ~50% of the smoothed intensity.

For the run-off experiments, the analysis is restricted to traces that start no later than 100 s after HT is added to the medium, so that translation is initially 'close' to the distribution in the absence of inhibitor, considering that HT takes ~60 s to enter the cells and be effective (*Ingolia et al., 2009*). Experiments with less than 10 traces in the control or in the perturbed (GC7/KO) conditions are excluded from the analysis. In addition, two acquisitions exhibited a delayed onset of translation, questioning the effectiveness of HT treatment in those instances and were also excluded.

## Mathematical models

### Bayesian modeling of run-off traces

We consider a set of intensity traces $y_{r,t}$, where $r$ is a trace index and $t$ is a discrete time index, which is a multiple of the time step $dt$. We assume a common underlying stochastic process that can be described by an initiation rate α and an average elongation rate $\lambda$ of the ribosomes along the transcript. In particular, we use a Hidden Markov Model (HMM) in which each trace $y_{r,t}$ is a noisy observation of a discrete Markov chain $N_{r,t}$, where $N_{r,t}$ is the number of translating ribosomes on the transcript $r$ at time $t$. The inference is performed in Stan using a Markov chain Monte Carlo algorithm called Hamiltonian Monte Carlo (HMC). It uses an approximate Hamiltonian dynamics simulation based on numerical integration which is then corrected by performing a Metropolis acceptance step (**Betancourt and Girolami, 2013**).

In the following, we describe how we derive the expressions of the initial state probability and the transition probabilities from a simple mean-field approximation of the $\ell$-TASEP under the assumption of low ribosome density. At the end of the section, we describe our assumptions related to the emission probability, that is the probability of observing the noisy signal $y$ given the hidden state $N$.

Given a transcript of size $L$, we define the average ribosome density $\rho$, defined as the average fraction of transcript occupied by the ribosomes

$$\rho := \frac{\langle N \rangle \ell}{L}, \tag{10}$$

where $\ell$ is the length of transcript occupied by one ribosome and $\langle N \rangle$ is the average number of ribosomes on the transcript. The latter can be expressed as the product between the current $J$ (number of particles per transcript site and per unit of time, equivalent to the protein synthesis rate) and the average residence time $L/\lambda$. Hence, **Equation 10** can be written as

$$\rho = J\frac{\ell}{\lambda}. \tag{11}$$

In the initiation-limited regime as described in the continuum-limit approximation by **Erdmann-Pham et al., 2020**, the current is mainly determined by the initiation rate and it is given by

$$J = \alpha\frac{\lambda - \alpha}{\lambda + (\ell - 1)\alpha}. \tag{12}$$

In our model, we assume that the initiation rate is low compared to elongation and ribosome excluded volume interactions can be neglected. In this case, we can expand the expression of $J$ as a function of the ratio $\alpha/\lambda$:

$$J = \alpha\left\{1 - \ell\frac{\alpha}{\lambda} + o\left[\left(\ell\frac{\alpha}{\lambda}\right)^2\right]\right\}, \tag{13}$$

to observe that when $\ell\frac{\alpha}{\lambda} \ll 1$, or equivalently $\lambda/\alpha \gg \ell$, we can approximate $J \approx \alpha$ and

$$\rho \approx \frac{\alpha}{\lambda}\ell \quad \text{if } \lambda/\alpha \gg \ell, \tag{14}$$

an approximation that we will use throughout our model. The initial state probability $\Theta_r(N)$ is the probability of observing $N$ ribosomes at the beginning of trace $r$, given the model parameters. We set $t = 0$ when the run-off starts, and we assume that this happens 60 s after HT is added to the medium (**Ingolia et al., 2011**). At $t = 0$ we can assume that the number of ribosomes $N$ on the transcript is given by a Poisson distribution with mean $\alpha L/\lambda$, as in the absence of inhibitor, and where the low-density approximation in **Equation 14** is assumed (**Szavits-Nossan and Grima, 2023**). Given that trace $r$ starts at time $t_r \geq 0$, we assume that the ribosome distribution has not yet been substantially altered by initiation inhibition, and we approximate it by a Poisson distribution with mean corrected by $t_r$

$$p_r(N|\alpha, \lambda) = \text{Poisson}\left(N\,\middle|\,\alpha\left(\frac{L}{\lambda} - t_r\right)\right). \tag{15}$$

However, the probability of having $N = 0$ ribosomes on the transcript, hence the probability that a transcript is inactive at the time of HT treatment, may not follow the Poisson distribution – for instance, in the presence of switching between active and inactive states (whether or not the Poisson distribution is still a valid description for $N > 0$ will depend on the switching rates; *Szavits-Nossan and Grima, 2023*). In practice, we describe the probability for a trace to be inactive as $p_{\text{off}}$, which is a parameter of the model and it is shared among all traces. Finally, the initial probability vector for trace $r$ is given by

$$\Theta_r(N \,|\, \alpha, \lambda, p_{\text{off}}) = \begin{cases} p_{\text{off}} & N = 0 \\ p_r(N) \,/\, (1 - p_r(0)) & N > 0. \end{cases} \tag{16}$$

By choosing such a Poisson distribution, we assume that, in case of bursting, the durations of active and inactive periods are longer than the total mRNA translation time. For the inference, we need to fix the maximum number of ribosomes $N_{\text{max}}$ that can be translating the mRNA. We fix $N_{\text{max}} = 100$, which corresponds to a density $\rho \approx 0.94$ for $L = 1066$, aa and $\ell = 10$ aa.

The transition probability $\Gamma(N \,|\, N', \alpha)$ is the probability of jumping from $N'$ to $N$ in a time $dt$, given the kinetic parameters α. Because at $t \geq 0$ initiation is inhibited, the only possible transitions are those satisfying $N \leq N'$, hence termination events.

When $\lambda/\alpha \gg \ell$ the waiting time distribution between successive termination events was derived in *Szavits-Nossan and Grima, 2023* and reads

$$P_{\text{term}}(t) = \alpha \frac{\lambda - \alpha}{\lambda - 2\alpha} \left( e^{-\alpha t} - e^{-(\lambda - \alpha)t} \right). \tag{17}$$

Since we are assuming $\alpha \ll \lambda/\ell < \lambda$ we can say that $\frac{\lambda - \alpha}{\lambda - 2\alpha} \approx 1$ and $e^{-(\lambda - \alpha)t} \ll e^{-\alpha t}$. As a consequence, *Equation 17* is approximately equal to the waiting time distribution between initiation events:

$$P_{\text{term}}(t) \approx \alpha e^{-\alpha t}. \tag{18}$$

So, we can define the transition probability from $N$ to $N - k$ given α as

$$\Gamma(N - k \,|\, N, \alpha) = \text{Poisson}(k \,|\, \alpha dt). \tag{19}$$

For the inference, we need to set a maximum number of termination events $k_{\text{max}}$ per time step $dt$. Considering that the initiation rates measured in the literature with similar systems range from less than 1 ribosome per min up to 5 ribosomes per min (~0.08 s$^{-1}$; *Boersma et al., 2019*; *Livingston et al., 2023*), we fixed $k_{\text{max}} = 6$, since the probability of 6 initiation events in $dt$ with $\alpha = 0.08$ s$^{-1}$ is about 0.005.

We assume that the emission probability $\Phi(y \,|\, N, t)$ – the probability of observing an intensity $y$ given $N$ ribosomes at time $t$ – is lognormal

$$\Phi\left(y \,|\, N, t\right) = \frac{1}{y\, s(t) \sqrt{2\pi}} \exp\left\{ -\frac{\left[\ln y - \ln \mu(N, t)\right]^2}{2s^2(t)} \right\} \tag{20}$$

to account for different sources of noise arising during acquisition and not filtered by the intensity quantification and pre-processing steps (*Wildner et al., 2023*). These sources include $z$-diffusion of the mRNA, irregular background, fluctuations in laser intensity, photon shot noise, etc. In addition, this choice aims to effectively describe the noise deriving from the uncertainty regarding the ribosome distribution along the mRNA. For a time $t$ and $N$ ribosomes, an equivalent representation in terms of random variables is

$$Y = \mu \exp\left(sX\right) \tag{21}$$

where $X$ is a standard normal random variable, $\mu := \mu(N, t)$ and $s := s(t)$. In practice, the measured spot intensity $y$ is given by the true signal μ multiplied by an exponentially transformed Gaussian noise.

The following discussion is dedicated to the derivation of $\mu(N, t)$ and $s(t)$ which explains the explicit dependence of the emission probability on time. Notice that the noise parameter $s$ can be expressed in terms of the expectation $\langle y(N, t) \rangle$ and variance $\sigma_y^2(N, t)$ of $y$ as

$$s^2(N, t) = \ln\left(1 + \frac{\sigma_y^2(N, t)}{\langle\, y(N, t)\,\rangle^2}\right). \tag{22}$$

The denoised signal μ is a function of the number of translating ribosomes $N$ and time $t$ and, since we are neglecting the precise ribosome positions along the mRNA, it is an average over all the possible configurations $\mathcal{C}_{N,t}$ of $N$ ribosomes along the mRNA at time $t$ of run-off. In practice, we consider that the ribosomes are uniformly distributed along the mRNA. We express it as

$$\mu(N, t) = \langle\, I(\mathcal{C}_{N,t})\,\rangle_{\mathcal{C}_{N,t}} = b_0 + \gamma(t) N i_{\mathrm{MP}}, \tag{23}$$

where $I(\mathcal{C}_{N,t})$ is the intensity generated by the ribosome configuration $\mathcal{C}_{N,t}$, $b_0$ is the offset, $i_{\mathrm{MP}}$ is the intensity of one mature protein and γ is a time-dependent correction factor that accounts for the fact that ribosomes translating the SunTag have intensity smaller than $i_{\mathrm{MP}}$. It depends on time because during run-off, the ensemble of possible ribosomes configurations along the mRNA changes, as ribosomes progress on the transcript. Given the length of the SunTag $L_S$, we can express γ(t) in terms of the elongation rate $\lambda$ (see Mean intensity correction factor during run-off for the full derivation):

$$\gamma(t) = 1 - \frac{(L_S - \lambda t)^2}{2 L_S (L - \lambda t)} \quad \text{for } \lambda t \leq L_S, \tag{24}$$

which agrees with the result for $t = 0$ previously reported (**Aguilera et al., 2019**). When all ribosomes have completed the translation of the SunTag, and approximately for $\lambda t > L_S$, γ(t) = 1. The value of γ over time agrees well with simulation results over a broad range of ribosome densities (**Figure 2C** and **Figure 2—figure supplement 2A**).

We can compute the variance of the total spot intensity in a similar way

$$\sigma^2(N, t) := \mathrm{Var}\left[I(\mathcal{C}_{N,t})\right]_{\mathcal{C}_{N,t}} = \nu(t) N i_{\mathrm{MP}}^2 \tag{25}$$

where $\nu$ is given by (see Intensity variance for the derivation)

$$\nu(t) = \frac{1}{L - \lambda t}\left(L - \frac{2 L_S}{3} - \frac{(\lambda t)^3}{3 L_S^2}\right) - \left(1 - \frac{(L_S - \lambda t)^2}{2 L_S (L - \lambda t)}\right)^2. \tag{26}$$

The agreement with simulations is good at low densities, while it significantly deteriorates at higher densities (**Figure 2D** and **Figure 2—figure supplement 2A**). This is expected as the expressions of both γ(t) and ν(t) are derived under the assumption that the ribosome density is sufficiently low that excluded-volume interactions can be neglected, and the probability to find a ribosome at position $i$ can be approximated by $N/L$.

We can write the total variance $\sigma_y^2(N, t)$ of the signal $y(N, t)$ as the sum of the variance $\sigma_{\mathrm{meas}}^2$ due to measurement noise and variance $\sigma_{\mathcal{C}}^2(N, t)$ due to the ribosome configurations on the mRNA, and effectively subsume both of them under the log-normal noise. So, the observed intensity $y$ is a random variable because of the unknown measurement process and because of the unknown underlying ribosome distribution. Then, the normalized variance in the expression of $s$ (**Equation 22**) reads

$$\frac{\sigma_y^2(N, t)}{\langle\, y(N, t)\,\rangle^2} = \frac{\sigma_{\mathrm{meas}}^2(N, t)}{\langle\, y(N, t)\,\rangle^2} + \frac{\sigma_{\mathcal{C}}(N, t)^2}{\langle\, y(N, t)\,\rangle^2} \tag{27}$$

where the average is intended over the measurements and the configurations $\mathcal{C}_{N,t}$. We approximate the first term in the sum as

$$\frac{\sigma_{\mathrm{meas}}^2(N, t)}{\langle\, y(N, t)\,\rangle^2} \approx \frac{\sigma_{\mathrm{meas}}^2(N, t)}{\langle\, I(\mathcal{C}_{N,t})\,\rangle_{\mathrm{meas}}^2} \tag{28}$$

where the average on the right-hand side is performed over the measurements and where we assume that the right-hand side is a constant multiplicative noise that does not depend on the specific ribosome configuration, nor the intensity, and subsumes the uncertainty in the acquisition process. We approximate the second term in the sum as

$$\frac{\sigma_C^2(N,t)}{\langle y(N,t)\rangle^2} \approx \frac{\sigma^2(N,t)}{\langle I(\mathcal{C}_{N,t})\rangle_{\mathcal{C}_{N,t}}^2} = \frac{\sigma^2(N,t)}{\mu^2} = \frac{1}{N}\frac{\nu(t)}{\gamma^2(t)} \tag{29}$$

and, by substituting $N$ with the average $\langle N\rangle = \alpha L/\lambda$,

$$\frac{\sigma_C^2(N,t)}{\langle y(N,t)\rangle^2} \approx \frac{\lambda}{\alpha L}\frac{\nu(t)}{\gamma^2(t)}. \tag{30}$$

Finally, we can use *Equation 22* to write the final estimate for $s$

$$s(t) = \sqrt{\ln\left(1 + \frac{\sigma_{\text{meas}}^2}{\langle I\rangle_{\text{meas}}^2} + \frac{\lambda}{\alpha L}\frac{\nu(t)}{\gamma^2(t)}\right)}, \tag{31}$$

where we have omitted the dependence of $I$ on ribosome configurations $\mathcal{C}_{N,t}$ for simplicity. Notice that $s(t)$ depends solely on time and on the kinetic parameters α and $\lambda$. When there are no ribosomes on the transcript or when $\nu(t) = 0$ (all ribosomes have terminated the translation of the SunTag) the noise is simply given by

$$s_0 := \sqrt{\ln\left(1 + \frac{\sigma_{\text{meas}}^2}{\langle I\rangle_{\text{meas}}^2}\right)}. \tag{32}$$

Finally, the emission probability is given by

$$\Phi\left(y\,|\,N,t\right) = \frac{1}{ys(t)\sqrt{2\pi}}\exp\left\{-\frac{\left[\ln y - \ln\mu(N,t)\right]^2}{2s^2(t)}\right\} \tag{33}$$

where $\mu(N,t)$ is given by *Equation 23* and $s(t)$ is given by *Equation 31*.

For each experiment, we estimate the traces baseline $b_0$ in *Equation 23* and the term $\sigma_{\text{meas}}^2/\langle I\rangle^2$ in *Equation 31* independently of the other model parameters. The offset $b_0$ is estimated as the average intensity of the untranslated traces: because it is very consistent across experiments and conditions, we fix it to its average value across experiments, $b_0 = 6$ a.u.

The term $\sigma_{\text{meas}}^2/\langle I\rangle^2$ determines the noise parameter $s_0$ (*Equation 32*), which accounts for the measurement noise alone (and not for the uncertainty of the ribosome distribution). It is estimated independently for each experiment (all conditions together) by calculating the variance of untranslated traces $\sigma_{\text{meas}}^2$ divided by the mean intensity squared of the trace $\langle I\rangle^2$. The values obtained in this way correspond to $s_0$ ranging from 0.31 to 0.38, with mean 0.34 and standard deviation 0.02 – across experiments. In an independent estimate from raw CHX traces described in Preprocessing and data filtering, we obtain $s_0 = 0.36$, a value comparable to what was obtained for untranslated traces (*Figure 2—figure supplement 1A, B*).

## Mean intensity correction factor at $t = 0$

We compute here the correction factor γ for the total intensity of $N$ ribosomes used in Bayesian modeling of run-off traces, at $t = 0$. The number of epitopes $n$ on the SunTag as a function of the position $x$ can be approximated as

$$n(x) = \frac{n_S}{L_S - 1}(x - 1) \quad x \leq L_S \tag{34}$$

where $n_S = 24$ is the total number of epitopes encoded in the SunTag and $L_S$ is the SunTag length. For $x > L_S$ we have $n(x) = n_S$. In the following, we will consider the intensity normalized by $i_{\text{MP}}$ and use the fraction of epitopes $f(x) := n(x)/n_S$. We can compute the average intensity for $N = 1$ ribosomes translating the SunTag as

$$\mu_1 = \sum_{x=1}^{L} f(x)p(x) \tag{35}$$

where $p(x)$ is the probability of finding a ribosome at $x$. After approximating $p(x) = 1/L$ and using the definition of $f(x)$ we have

$$\mu_1 = \frac{1}{L}\sum_{x=1}^{L} f(x) = \frac{1}{L}\left[\frac{1}{L_S - 1}\sum_{x=1}^{L_S}(x-1) + (L - L_S)\right]$$

which yields

$$\mu_1 = 1 - \frac{L_s}{2L} = \gamma, \tag{36}$$

where we used

$$\sum_{x=1}^{L_S}(x-1) = \frac{L_S(L_S - 1)}{2}.$$

For $N > 1$ we have $\mu_N = N\mu_1 = N\gamma$.

## Mean intensity correction factor during run-off

We compute here the correction factor $\gamma$ for the total intensity of $N$ ribosomes used in Bayesian modeling of run-off traces during run-off. While ribosomes run off the transcripts, the proportion of ribosomes on the SunTag changes, which can be taken into account by a time-dependent correction factor $\gamma(t)$, dependent on the elongation rate $\lambda$ (that we assume to be uniform along the transcript). We can approximate the occupation probability as

$$p(x\,|\,N, t) = \begin{cases} 0 & \forall x \leq x_t \\ N/(L - x_t) & \forall x > x_t \end{cases} \tag{37}$$

where we defined $x_t := \lfloor \lambda t \rfloor \in [1, L]$, the average distance covered by the ribosomes in a time $t$. With the same algebra as before, we get $\mu_N = N\gamma(t)$ with

$$\gamma(t) = \frac{1}{L - x_t}\left(L - \frac{L_S}{2} - \frac{x_t(x_t - 1)}{2(L_S - 1)}\right) \simeq 1 - \frac{(L_S - \lambda t)^2}{2L_S(L - \lambda t)} \quad \forall t \leq L_S/\lambda \tag{38}$$

else $\gamma(t) = 1$.

## Intensity variance

We compute the variance $V_1(y)$ of the intensity $y$ produced by $N = 1$ ribosomes at time $t$ during run-off, in units of $i_{\mathrm{MP}}^2$ ($t = 0$ corresponds to when run-off starts). The variance for $N$ ribosomes will be the sum of the variances: $V_N(y) = NV_1(y)$. We can write the variance as

$$V_1(y) = \left\langle y^2 \right\rangle - \mu_1^2 \tag{39}$$

with $\mu_1$ given by *Equation 36* and

$$\left\langle y^2 \right\rangle = \sum_{x=x_t}^{L} f(x)^2 p(x) = \frac{1}{L - x_t}\left(\sum_{x=1}^{L} f(x)^2 - \sum_{x=1}^{x_t} f(x)^2\right) \tag{40}$$

where $x_t := \lfloor \lambda t \rfloor \in [1, L]$ as before. The first summation gives

$$\sum_{x=1}^{L} f(x)^2 = \frac{1}{(L_S - 1)^2}\sum_{x=1}^{L_S}(x-1)^2 + L - L_S$$

$$= \frac{1}{(L_S - 1)^2}\left(\frac{L_S(L_S + 1)(2L_S + 1)}{6} - L_S(L_S + 1) + L_S\right) + L - L_S$$

$$= \frac{L_S(2L_S - 1)}{6(L_S - 1)} + L - L_S \approx L - \frac{2L_S}{3}.$$

The second term

$$\sum_{x=1}^{x_t} f(x)^2 = \frac{1}{(L_S - 1)^2} \sum_{x=1}^{x_t} (x - 1)^2$$

$$= \frac{1}{(L_S - 1)^2} \left( \frac{x_t(x_t + 1)(2x_t + 1)}{6} - x_t(x_t + 1) + x_t \right)$$

$$= \frac{x_t(2x_t - 1)(x_t - 1)}{6(L_S - 1)^2} \approx \frac{(\lambda t)^3}{3L_S^2}.$$

Finally,

$$\left\langle y^2 \right\rangle \approx \frac{1}{L - \lambda t} \left( L - \frac{2L_S}{3} - \frac{(\lambda t)^3}{3L_S^2} \right).$$

The final results are then

$$V_1(y) = \frac{1}{L - \lambda t} \left( L - \frac{2L_S}{3} - \frac{(\lambda t)^3}{3L_S^2} \right) - \left( 1 - \frac{(L_S - \lambda t)^2}{2L_S(L - \lambda t)} \right)^2. \tag{41}$$

## Simulation of SunTag traces

SunTag traces are obtained in silico by simulating the homogeneous $\ell$-TASEP with the Gillespie algorithm (*Gillespie, 1976*). The algorithm is formulated for a general set of site-specific elongation rates $\lambda_i$, but for the purposes of this study, we simulate a homogeneous system with all elongation rates equal to $\lambda$; for simplicity, the termination rate $\lambda_{L-1}$ is considered equal to the elongation rate. We first describe the algorithm used for the $\ell$-TASEP simulations without bursting, and then we describe its generalization for a bursting initiation.

The simulation starts at $t = 0$. At any step of the simulation, a vector $\tau = (\tau_0, \ldots, \tau_{L-1}) \in \{0,1\}^L$ specifies the ribosome occupancy state of the mRNA (of length $L$) and a list $\Omega$ specifies all the possible reactions and their associated time. In particular, $\omega_k \in \Omega$ is a list of two elements: $\omega_k[0] \in \{0, \ldots, L-1\}$ specifies the event and $\omega_k[1] \in \mathbb{R}^+$ its time of occurrence. $\omega_k[0] = -1$ corresponds to an initiation event, $\omega_k[0] = L - 1$ a termination event, and $\omega_k[0] \in \{0, \ldots, L-2\}$ an elongation event at position $\omega_k[0]$. The list $\Omega$ is kept ordered according to the time of occurrence, such that the first element of the list is the one occurring earlier in time and $\omega_k[1] \leq \omega_{k+1}[1]$. In the following $\mathrm{Exp}(\lambda)$ is the exponential distribution with scale parameter $\lambda$.

1. Remove the event $\omega_1$ from $\Omega$ (let $i = \omega_1[0]$ and $dt = \omega_1[1]$).
2. Update the total time $t \leftarrow t + dt$ and the single events time $\omega_k[1] \leftarrow \omega_k[1] - dt \,\forall k$.
3. Update the occupancy state:
   a. If $i = -1$ (initiation), then set $\tau_0 = 1$.
   b. If $i = L - 1$ (termination), then set $\tau_{L-1} = 0$.
   c. If $i \in \{0, \ldots, L-2\}$ (elongation), then set $(\tau_i, \tau_{i+1}) \leftarrow (0, 1)$.
4. Find possible new reactions and sample their associated times:
   a. If $i = L - 1$ and $\tau_{L-\ell-1} = 1$ then sample $t_0 \sim \mathrm{Exp}(\lambda_{L-\ell-1})$ and insert $(L - \ell - 1, t_0)$ into $\Omega$.
   b. If $i \in \{0, \ldots, L-2\}$
      i. If $i = \ell - 1$, then sample $t_0 \sim \mathrm{Exp}(\alpha)$ and insert $(-1, t_0)$ into $\Omega$.
      ii. If $i > \ell - 1$ and $\tau_{i-\ell} = 1$, then sample $t_0 \sim \mathrm{Exp}(\lambda_{i-\ell})$ and insert $(i - \ell, t_0)$ into $\Omega$.
      iii. If $i < L - \ell$ and $\tau_{i+\ell+1} = 0$ or if $L - \ell \leq i \leq L - 2$, then sample $t_0 \sim \mathrm{Exp}(\lambda_{i+1})$ and insert $(i + 1, t_0)$ into $\Omega$.
5. Go to 1.

The algorithm easily generalizes to a two-state telegraph model for initiation: the state of the system is now described by $\tau$ and $G \in \{0, 1\}$, with $G = 0$ if the transcript is inactive and $G = 1$ if the transcript is active. In the inactive state, initiation is not possible, while in the active state, initiation is

an exponential process with rate α. In practice, we add the reaction $\omega_k[0] = -2$ that corresponds to a change in transcript activity, $G \rightarrow \neg G$. The reaction time is drawn from an exponential distribution

$$\omega_k[1] \sim \begin{cases} \text{Exp}(k_1) & \text{if } G = 0 \\ \text{Exp}(k_0) & \text{if } G = 1. \end{cases}$$

We first simulate the model for $10^3$ iterations of burn-in; for each trace, we record the state of the system every $dt = 20$ s for a total duration of 90 min. The final state is used as a starting point to simulate the dynamics for $10^3$ iterations before recording the following trace. For the run-off, we remove the initiation events possibly stored in $\Omega$ after the initial burn-in and before simulating the first trace with $\alpha = 0$. The last state of burn-in is then used as a starting point to simulate the dynamics for $10^3$ more iterations before recording the following run-off.

To obtain the translation signal, we compute the number of epitopes $n(t)$ synthesized at each time point $t$ by all the translating ribosomes. This is given by the scalar product

$$n(t) = \boldsymbol{w} \cdot \boldsymbol{\tau}(t) \tag{42}$$

where $\boldsymbol{w} \in \mathbb{N}^L$, with $L$ the transcript size, containing the cumulative number of epitopes along the transcript, and $\boldsymbol{\tau} \in \{0,1\}^L$ is the time-dependent occupation vector, with

$$\tau_i = \begin{cases} 1 & \text{if site } i \text{ is occupied} \\ 0 & \text{if site } i \text{ is free.} \end{cases} \tag{43}$$

The total intensity is given by $I_S(t) = \frac{i_{\text{MP}}}{24} n(t)$, where $i_{\text{MP}}$ is the intensity of one mature protein and the suffix S stands for 'simulated'.

We add an offset $b_0$ to the intensity and we apply a log-normal noise to obtain the final simulated, noisy signal $y_S(t)$:

$$P(y_S(t) \,|\, \mu_S(t), s_0) = \frac{1}{y_S(t) s_0 \sqrt{2\pi}} \exp \left\{ -\frac{(\ln y_S(t) - \ln \mu_S(t))^2}{2 s_0^2} \right\} \tag{44}$$

where $\mu_S(t) = b_0 + I_S(t)$ and $s_0$ is the measurement noise, as described at the end of Bayesian modeling of run-off traces.

The duration of each trace is drawn from an exponential distribution with mean 10 min to mimic the tracking duration of one mRNA molecule in experiments. Simulated run-off traces are longer than 5 min (as in experiments, see Spot tracking and intensity quantification) and start at $t = 0$, when the run-off starts (approximately 60 s after HT addition, see Preprocessing and data filtering).

## Acknowledgements

We would like to thank the EPFL's Bioimaging and Optics Facility (BIOP) and Scientific IT & Application Support (SCITAS) for their assistance, as well as Eva Poch for her help with the Western blot experiments. We also appreciate the insightful discussions with Tobias Hochstoeger regarding the SunTag system. This research was supported by the Swiss National Science Foundation (SNSF) Sinergia grant #205884 to FN and JC, the Swiss Cancer Research grant #KFS-5412-08-2021-R to FN,and the École Polytechnique Fédérale de Lausanne.

# Additional information

## Funding

| Funder | Grant reference number | Author |
|---|---|---|
| Swiss National Science Foundation | 205884 | Jeffrey A Chao Felix Naef |
| Swiss Cancer Research Grant | KFS-5412-08-2021-R | Felix Naef |

The funders had no role in study design, data collection and interpretation, or the decision to submit the work for publication.

## Author contributions

Irene Lamberti, Conceptualization, Resources, Data curation, Formal analysis, Investigation, Visualization, Methodology, Writing – original draft, Writing – review and editing; Jeffrey A Chao, Resources, Supervision, Methodology; Cédric Gobet, Conceptualization, Data curation, Supervision, Investigation, Methodology, Writing – original draft, Writing – review and editing; Felix Naef, Conceptualization, Resources, Supervision, Methodology, Writing – original draft, Writing – review and editing

## Author ORCIDs

Cédric Gobet ⬤ https://orcid.org/0000-0001-5627-3444
Felix Naef ⬤ https://orcid.org/0000-0001-9786-3037

Reviewer #1 (Public review): https://doi.org/10.7554/eLife.107160.3.sa1
Reviewer #2 (Public review): https://doi.org/10.7554/eLife.107160.3.sa2
Reviewer #3 (Public review): https://doi.org/10.7554/eLife.107160.3.sa3
Author response https://doi.org/10.7554/eLife.107160.3.sa4

# Additional files

## Supplementary files

MDAR checklist

Supplementary file 1. Nucleotide sequences of the COL1A1 and mutated COL1A1 (proline-to-alanine) inserts used in the PPG and AAG SunTag reporters, respectively.

## Data availability

Our computational models for analysis of SunTag traces are available on Github (https://github.com/naef-lab/suntag-analysis copy archived at *Lamberti, 2026*). SunTag run-off and steady-state raw images are available on Zenodo (https://doi.org/10.5281/zenodo.17669331).

The following dataset was generated:

| Author(s) | Year | Dataset title | Dataset URL | Database and Identifier |
|---|---|---|---|---|
| Lamberti I, Chao JA, Gobet C, Naef F | 2025 | SunTag run-off and steady-state raw images from Lamberti et al., 2025 | https://doi.org/10.5281/zenodo.17669331 | Zenodo, 10.5281/zenodo.17669331 |

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
