## [Editor Report · eLife Assessment]

This **important** study provides evidence for dynamic coupling between translation initiation and elongation that can help maintain low ribosome density and translational homeostasis. The authors combine single-molecule imaging with a new approach to analyze mRNA translation kinetics using Bayesian modeling. This work is overall **solid** and will be of interest to those studying translational regulation.

---

## [Referee Report · Reviewer #1 (Public review)]

Summary:

In this study, Lamberti et al. investigate how translation initiation and elongation are coordinated at the single-mRNA level in mammalian cells. The authors aim to uncover whether and how cells dynamically adjust initiation rates in response to elongation dynamics, with the overarching goal of understanding how translational homeostasis is maintained. To this end, the study combines single-molecule live-cell imaging using the SunTag system with a kinetic modeling framework grounded in the Totally Asymmetric Simple Exclusion Process (TASEP). By applying this approach to custom reporter constructs with different coding sequences, and under perturbations of the initiation/elongation factor eIF5A, the authors infer initiation and elongation rates from individual mRNAs and examine how these rates covary.

The central finding is that initiation and elongation rates are strongly correlated across a range of coding sequences, resulting in consistently low ribosome density ({less than or equal to}12% of the coding sequence occupied). This coupling is preserved under partial pharmacological inhibition of eIF5A, which slows elongation but is matched by a proportional decrease in initiation, thereby maintaining ribosome density. However, a complete genetic knockout of eIF5A disrupts this coordination, leading to reduced ribosome density, potentially due to changes in ribosome stalling resolution or degradation.

Strengths:

A key strength of this work is its methodological innovation. The authors develop and validate a TASEP-based Hidden Markov Model (HMM) to infer translation kinetics at single-mRNA resolution. This approach provides a substantial advance over previous population-level or averaged models and enables dynamic reconstruction of ribosome behavior from experimental traces. The model is carefully benchmarked against simulated data and appropriately applied. The experimental design is also strong. The authors construct matched SunTag reporters differing only in codon composition in a defined region of the coding sequence, allowing them to isolate the effects of elongation-related features while controlling for other regulatory elements. The use of both pharmacological and genetic perturbations of eIF5A adds robustness and depth to the biological conclusions. The results are compelling: across all constructs and conditions, ribosome density remains low, and initiation and elongation appear tightly coordinated, suggesting an intrinsic feedback mechanism in translational regulation. These findings challenge the classical view of translation initiation as the sole rate-limiting step and provide new insights into how cells may dynamically maintain translation efficiency and avoid ribosome collisions.

Assessment of Goals and Conclusions:

The authors successfully achieve their stated aims: they quantify translation initiation and elongation at the single-mRNA level and show that these processes are dynamically coupled to maintain low ribosome density. The modeling framework is well suited to this task, and the conclusions are supported by multiple lines of evidence, including inferred kinetic parameters, independent ribosome counts, and consistent behavior under perturbation.

Impact and Utility:

This work makes a significant conceptual and technical contribution to the field of translation biology. The modeling framework developed here opens the door to more detailed and quantitative studies of ribosome dynamics on single mRNAs and could be adapted to other imaging systems or perturbations. The discovery of initiation-elongation coupling as a general feature of translation in mammalian cells will likely influence how researchers think about translational regulation under homeostatic and stress conditions.

The data, models, and tools developed in this study will be of broad utility to the community, particularly for researchers studying translation dynamics, ribosome behavior, or the effects of codon usage and mRNA structure on protein synthesis.

Context and Interpretation:

This study contributes to a growing body of evidence that translation is not merely controlled at initiation but involves feedback between elongation and initiation. It supports the emerging view that ribosome collisions, stalling, and quality control pathways play active roles in regulating initiation rates in cis. The findings are consistent with recent studies in yeast and metazoans showing translation initiation repression following stalling events. However, the mechanistic details of this feedback remain incompletely understood and merit further investigation, particularly in physiological or stress contexts.

In summary, this is a thoughtfully executed and timely study that provides valuable insights into the dynamic regulation of translation and introduces a modeling framework with broad applicability. It will be of interest to a wide audience in molecular biology, systems biology, and quantitative imaging.

---

## [Referee Report · Reviewer #2 (Public review)]

Summary:

This manuscript uses single-molecule run-off experiments and TASEP/HMM models to estimate biophysical parameters, i.e., ribosomal initiation and elongation rates. Combining inferred initiation and elongation rates, the authors quantify ribosomal density. TASEP modeling was used to simulate the mechanistic dynamics of ribosomal translation, and the HMM is used to link ribosomal dynamics to microscope intensity measurements. The authors' main conclusions and findings are:

- Ribosomal elongation rates and initiation rates are strongly coordinated.

- Elongation rates were estimated between 1 and 4.5 aa/sec. Initiation rates were estimated between 1 and 2 ribosomes/min. These values agree with previously reported ones.

- Ribosomal density was determined to be below 12% for all constructs and conditions.

- eIF5A-perturbations (GC7 inhibition) resulted in non-significant changes in translational bursting and ribosome density.

- eIF5A perturbations affected both elongation and initiation rates.

Strengths:

This manuscript presents an interesting scientific hypothesis to study ribosome initiation and elongation concurrently. This topic is relevant for the field. The manuscript presents a novel quantitative methodology to estimate ribosomal initiation rates from Harringtonine run-off assays. This is relevant because run-off assays have been used to estimate, exclusively, elongation rates.

Comments on revisions:

The authors have addressed my concerns. Specifically, they have expanded the discussion on unexpected eIF5A perturbation results, calculated CAI values for all constructs, and made code and data publicly available via GitHub and Zenodo. The mathematical notation is now consistent, and all variables are properly defined.

---

## [Referee Report · Reviewer #3 (Public review)]

Disclaimer:

My expertise is in live single-molecule imaging of RNA and transcription, as well as associated data analysis and modeling. While this aligns well with the technical aspects of the manuscript, my background in translation is more limited, and I am not best positioned to assess the novelty of the biological conclusions.

Summary:

This study combines live-cell imaging of nascent proteins on single mRNAs with time-series analysis to investigate the kinetics of mRNA translation.

The authors (i) used a calibration method for estimating absolute ribosome counts, and (ii) developed a new Bayesian approach to infer ribosome counts over time from run-off experiments, enabling estimation of elongation rates and ribosome density across conditions.

They report (i) translational bursting at the single-mRNA level, (ii) low ribosome density (~10% occupancy {plus minus} a few percents), (iii) that ribosome density is minimally affected by perturbations of elongation (using a drug and/or different coding sequences in the reporter), suggesting a homeostatic mechanism potentially involving a feedback of elongation onto initiation, although (iv) this coupling breaks down upon knockout of elongation factor eIF5A.

Strengths:

(1) The manuscript is well written and the conclusions are in general appropriately cautious (besides the few improvements I suggest below).

(2) The time-series inference method is interesting and promising for broader application.

(3) Simulations provide convincing support for the modeling (though some improvements are possible).

(4) The reported homeostatic effect on ribosome density is surprising and carefully validated with multiple perturbations.

(5) Imaging quality and corrections (e.g., flat-fielding, laser power measurements) are robust.

(6) Mathematical modeling is clearly described and precise; a few clarifications could improve it further.

Weaknesses:

(1) The absolute quantification of ribosome numbers (via the measurement of $i_{MP}$) should be improved. This only affects the finding that ribosome density is low, not that it appears to be under homeostatic control. However, if $i_{MP}$ turns out to be substantially overestimated (hence ribosome density underestimated), then "ribosomes queuing up to the initiation site and physically blocking initiation" could become a relevant hypothesis. In my first review of this work, I made recommendations, which the authors did not follow. In my view, the robustness of this particular aspect of this study remains moderate.

(2) The proposed initiation-elongation coupling is plausible, but alternative explanations such as changes in abortive elongation frequency should be considered. In their response to my previous comments, the authors indicate that this is "beyond the scope of the present work".

(3) More an opportunity for improvement than a weakness: It is unclear what the single-mRNA nature of the inference method is bringing since it is only used here to report _average_ ribosome elongation rate and density (averaged across mRNAs and across time during the run-off experiments -although the method, in principle, has the power to resolve these two aspects). In response to my previous comment, the authors note that such analyses could be incorporated in future work.

---

## [Author Response]

The following is the authors’ response to the original reviews.

**Public Reviews:**

**Reviewer #1 (Public review):**
Summary:In this study, Lamberti et al. investigate how translation initiation and elongation are coordinated at the single-mRNA level in mammalian cells. The authors aim to uncover whether and how cells dynamically adjust initiation rates in response to elongation dynamics, with the overarching goal of understanding how translational homeostasis is maintained. To this end, the study combines single-molecule live-cell imaging using the SunTag system with a kinetic modeling framework grounded in the Totally Asymmetric Simple Exclusion Process (TASEP). By applying this approach to custom reporter constructs with different coding sequences, and under perturbations of the initiation/elongation factor eIF5A, the authors infer initiation and elongation rates from individual mRNAs and examine how these rates covary.The central finding is that initiation and elongation rates are strongly correlated across a range of coding sequences, resulting in consistently low ribosome density ({less than or equal to}12% of the coding sequence occupied). This coupling is preserved under partial pharmacological inhibition of eIF5A, which slows elongation but is matched by a proportional decrease in initiation, thereby maintaining ribosome density. However, a complete genetic knockout of eIF5A disrupts this coordination, leading to reduced ribosome density, potentially due to changes in ribosome stalling resolution or degradation.Strengths:A key strength of this work is its methodological innovation. The authors develop and validate a TASEP-based Hidden Markov Model (HMM) to infer translation kinetics at single-mRNA resolution. This approach provides a substantial advance over previous population-level or averaged models and enables dynamic reconstruction of ribosome behavior from experimental traces. The model is carefully benchmarked against simulated data and appropriately applied. The experimental design is also strong. The authors construct matched SunTag reporters differing only in codon composition in a defined region of the coding sequence, allowing them to isolate the effects of elongation-related features while controlling for other regulatory elements. The use of both pharmacological and genetic perturbations of eIF5A adds robustness and depth to the biological conclusions. The results are compelling: across all constructs and conditions, ribosome density remains low, and initiation and elongation appear tightly coordinated, suggesting an intrinsic feedback mechanism in translational regulation. These findings challenge the classical view of translation initiation as the sole rate-limiting step and provide new insights into how cells may dynamically maintain translation efficiency and avoid ribosome collisions.

We thank the reviewer for their constructive assessment of our work, and for recognizing the methodological innovation and experimental rigor of our study.

Weaknesses:A limitation of the study is its reliance on exogenous reporter mRNAs in HeLa cells, which may not fully capture the complexity of endogenous translation regulation. While the authors acknowledge this, it remains unclear how generalizable the observed coupling is to native mRNAs or in different cellular contexts.

We agree that the use of exogenous reporters is a limitation inherent to the SunTag system, for which there is currently no simple alternative for single-mRNA translation imaging. However, we believe our findings are likely generalizable for several reasons.

As discussed in our introduction and discussion, there is growing mechanistic evidence in the literature for coupling between elongation (ribosome collisions) and initiation via pathways such as the GIGYF2-4EHP axis (Amaya et al. 2018, Hickey et al. 2020, Juszkiewicz et al. 2020), which might operate on both exogenous and endogenous mRNAs.

As already acknowledged in our limitations section, our exogenous reporters may not fully recapitulate certain aspects of endogenous translation (e.g., ER-coupled collagen processing), yet the observed initiation-elongation coupling was robust across all tested constructs and conditions.

We have now expanded the Discussion (L393-395) to cite complementary evidence from Dufourt et al. (2021), who used a CRISPR-based approach in Drosophila embryos to measure translation of endogenous genes. We also added a reference to Choi et al. 2025, who uses a ER-specific SunTag reporter to visualize translation at the ER (L395-397).

Additionally, the model assumes homogeneous elongation rates and does not explicitly account for ribosome pausing or collisions, which could affect inference accuracy, particularly in constructs designed to induce stalling. While the model is validated under low-density assumptions, more work may be needed to understand how deviations from these assumptions affect parameter estimates in real data.

We agree with the reviewer that the assumption of homogeneous elongation rates is a simplification, and that our work represents a first step towards rigorous single-trace analysis of translation dynamics. We have explicitly tested the robustness of our model to violations of the low-density assumption through simulations (Figure 2 - figure supplement 2). These show that while parameter inference remains accurate at low ribosome densities, accuracy slightly deteriorates at higher densities, as expected. In fact, our experimental data do provide evidence for heterogeneous elongation: the waiting times between termination events deviate significantly from an exponential distribution (Figure 3 - figure supplement 2C), indicating the presence of ribosome stalling and/or bursting, consistent with the reviewer's concern. We acknowledge in the Limitations section (L402-406) that extending the model to explicitly capture transcript-dependent elongation rates and ribosome interactions remains challenging. The TASEP is difficult to solve analytically under these conditions, but we note that simulation-based inference approaches, such as particle filters to replace HMMs, could provide a path forward for future work to capture this complexity at the single-trace level.

Furthermore, although the study observes translation "bursting" behavior, this is not explicitly modeled. Given the growing recognition of translational bursting as a regulatory feature, incorporating or quantifying this behavior more rigorously could strengthen the work's impact.

While we do not explicitly model the bursting dynamics in the HMM framework, we have quantified bursting behavior directly from the data. Specifically, we measure the duration of translated (ON) and untranslated (OFF) periods across all reporters and conditions (Figure 1G for control conditions and Figure 4G-H for perturbed conditions), finding that active translation typically lasts 10-15 minutes interspersed with shorter silent periods of 5-10 minutes. This empirical characterization demonstrates that bursting is a consistent feature of translation across our experimental conditions. The average duration of silent periods is similar to what was inferred by Livingston et al. 2023 for a similar SunTag reporter; while the average duration of active periods is substantially shorter (~15 min instead of ~40 min), which is consistent with the shorter trace duration in our system compared to theirs (~15 min compared to ~80 min, on average). Incorporating an explicit two-state or multi-state bursting model into the TASEP-HMM framework would indeed be computationally intensive and represents an important direction for future work, as it would enable inference of switching rates alongside initiation and elongation parameters. We have added this point to the Discussion (L415-417).

Assessment of Goals and Conclusions:The authors successfully achieve their stated aims: they quantify translation initiation and elongation at the single-mRNA level and show that these processes are dynamically coupled to maintain low ribosome density. The modeling framework is well suited to this task, and the conclusions are supported by multiple lines of evidence, including inferred kinetic parameters, independent ribosome counts, and consistent behavior under perturbation.Impact and Utility:This work makes a significant conceptual and technical contribution to the field of translation biology. The modeling framework developed here opens the door to more detailed and quantitative studies of ribosome dynamics on single mRNAs and could be adapted to other imaging systems or perturbations. The discovery of initiation-elongation coupling as a general feature of translation in mammalian cells will likely influence how researchers think about translational regulation under homeostatic and stress conditions.The data, models, and tools developed in this study will be of broad utility to the community, particularly for researchers studying translation dynamics, ribosome behavior, or the effects of codon usage and mRNA structure on protein synthesis.Context and Interpretation:This study contributes to a growing body of evidence that translation is not merely controlled at initiation but involves feedback between elongation and initiation. It supports the emerging view that ribosome collisions, stalling, and quality control pathways play active roles in regulating initiation rates in cis. The findings are consistent with recent studies in yeast and metazoans showing translation initiation repression following stalling events. However, the mechanistic details of this feedback remain incompletely understood and merit further investigation, particularly in physiological or stress contexts.In summary, this is a thoughtfully executed and timely study that provides valuable insights into the dynamic regulation of translation and introduces a modeling framework with broad applicability. It will be of interest to a wide audience in molecular biology, systems biology, and quantitative imaging.

We appreciate the reviewer's thorough and positive assessment of our work, and that they recognize both the technical innovation of our modeling framework and its potential broad utility to the translation biology community. We agree that further mechanistic investigation of initiation-elongation feedback under various physiological contexts represents an important direction for future research.

**Reviewer #2 (Public review):**
Summary:This manuscript uses single-molecule run-off experiments and TASEP/HMM models to estimate biophysical parameters, i.e., ribosomal initiation and elongation rates. Combining inferred initiation and elongation rates, the authors quantify ribosomal density. TASEP modeling was used to simulate the mechanistic dynamics of ribosomal translation, and the HMM is used to link ribosomal dynamics to microscope intensity measurements. The authors' main conclusions and findings are:(1) Ribosomal elongation rates and initiation rates are strongly coordinated.(2) Elongation rates were estimated between 1-4.5 aa/sec. Initiation rates were estimated between 0.5-2.5 events/min. These values agree with previously reported values.(3) Ribosomal density was determined below 12% for all constructs and conditions.(4) eIF5A-perturbations (KO and GC7 inhibition) resulted in non-significant changes in translational bursting and ribosome density.(5) eIF5A perturbations resulted in increases in elongation and decreases in initiation rates.Strengths:This manuscript presents an interesting scientific hypothesis to study ribosome initiation and elongation concurrently. This topic is highly relevant for the field. The manuscript presents a novel quantitative methodology to estimate ribosomal initiation rates from Harringtonine run-off assays. This is relevant because run-off assays have been used to estimate, exclusively, elongation rates.

We thank the reviewer for their careful evaluation of our work and for recognizing the novelty of our quantitative methodology to extract both initiation and elongation rates from harringtonine run-off assays, extending beyond the traditional use of these experiments.

Weaknesses:The conclusion of the strong coordination between initiation and elongation rates is interesting, but some results are unexpected, and further experimental validation is needed to ensure this coordination is valid.

We agree that some of our findings need further experimental investigation in future studies. However, we believe that the coordination between initiation and elongation is supported by multiple results in our current work: (1) the strong correlation observed across all reporters and conditions (Figure 3E), and (2) the consistent maintenance of low ribosome density despite varying elongation rates. While additional experimental validation would be valuable, we note that directly manipulating initiation or elongation independently in mammalian cells remains technically challenging. Nevertheless, our findings are consistent with emerging mechanistic understanding of collision-sensing pathways (GIGYF2-4EHP) that could mediate such coupling, as discussed in our manuscript.

(1) eIF5a perturbations resulted in a non-significant effect on the fraction of translating mRNA, translation duration, and bursting periods. Given the central role of eIF5a, I would have expected a different outcome. I would recommend that the authors expand the discussion and review more literature to justify these findings.

We appreciate this comment. This finding is indeed discussed in detail in our manuscript (Discussion, paragraphs 6-7). As we note there, while eIF5A plays a critical role in elongation, the maintenance of bursting dynamics and ribosome density upon perturbation can be explained by compensatory feedback mechanisms. Specifically, the coordinated decrease in initiation rates that counterbalances slower elongation to maintain homeostatic ribosome density. We also discuss several factors that complicate interpretation: (1) potential RQC-mediated degradation masking stronger effects in proline-rich constructs, (2) differences between GC7 treatment and genetic knockout suggesting altered stalling resolution kinetics, and (3) the limitations of using exogenous reporters that lack ER-coupled processing, which may be critical for eIF5A function in endogenous collagen translation (as suggested by Rossi et al., 2014; Mandal et al., 2016; Barba-Aliaga et al., 2021). The mechanistic complexity and tissue-specific nature of eIF5A function in mammals, which differs substantially from the better-characterized yeast system, likely contributes to the nuanced phenotype we observe. We believe our Discussion adequately addresses these points.

(2) The AAG construct leading to slow elongation is very surprising. It is the opposite of the field consensus, where codon-optimized gene sequences are expected to elongate faster. More information about each construct should be provided. I would recommend more bioinformatic analysis on this, for example, calculating CAI for all constructs, or predicting the structures of the proteins.

We agree that the slow elongation of the AAG construct is counterintuitive and indeed surprising. Following the reviewer's suggestion, we have now calculated the Codon Adaptation Index (CAI) for all constructs (Renilla 0.89, Col1a1 0.78, Col1a1 mutated 0.74). It is therefore unlikely that codon bias explains the slow translation, particularly since we designed the mutated Col1a1 construct with alanine codons selected to respect human codon usage bias, thereby minimizing changes in codon optimality. As we discuss in the manuscript, we hypothesize that the proline-to-alanine substitutions disrupted co-translational folding of the collagen-derived sequence. Prolines are critical for collagen triple-helix formation (Shoulders and Raines, 2009), and their replacement with alanines likely generates misfolded intermediates that cause ribosome stalling (Barba-Aliaga et al., 2021; Komar et al., 2024). This interpretation is supported by the high frequency (>30%) of incomplete run-off traces for AAG, suggesting persistent stalling events. Our findings thus illustrate an important potential caveat: "optimizing" a sequence based solely on codon usage can be detrimental when it disrupts functionally important structural features or co-translational folding pathways.

This highlights that elongation rates depend not only on codon optimality but also on the interplay between nascent chain properties and ribosome progression.

(3) The authors should consider using their methodology to study the effects of modifying the 5'UTR, resulting in changes in initiation rate and bursting, such as previously shown in reference Livingston et al., 2023. This may be outside of the scope of this project, but the authors could add this as a future direction and discuss if this may corroborate their conclusions.

We thank the reviewer for this excellent suggestion. We agree that applying our methodology to 5'-UTR variants would provide a complementary test of initiation-elongation coupling, and we have now added this as a future direction in the Discussion (L417-420).

(4) The mathematical model and parameter inference routines are central to the conclusions of this manuscript. In order to support reproducibility, the computational code should be made available and well-documented, with a requirements file indicating the dependencies and their versions.

We have added the Github link in the manuscript (https://github.com/naef-lab/suntag-analysis) and have also deposited the data (.ome.tif) on Zenodo (https://zenodo.org/records/17669332).

**Reviewer #3 (Public review):**
Disclaimer:My expertise is in live single-molecule imaging of RNA and transcription, as well as associated data analysis and modeling. While this aligns well with the technical aspects of the manuscript, my background in translation is more limited, and I am not best positioned to assess the novelty of the biological conclusions.Summary:This study combines live-cell imaging of nascent proteins on single mRNAs with time-series analysis to investigate the kinetics of mRNA translation.The authors (i) used a calibration method for estimating absolute ribosome counts, and (ii) developed a new Bayesian approach to infer ribosome counts over time from run-off experiments, enabling estimation of elongation rates and ribosome density across conditions.They report (i) translational bursting at the single-mRNA level, (ii) low ribosome density (~10% occupancy {plus minus} a few percents), (iii) that ribosome density is minimally affected by perturbations of elongation (using a drug and/or different coding sequences in the reporter), suggesting a homeostatic mechanism potentially involving a feedback of elongation onto initiation, although (iv) this coupling breaks down upon knockout of elongation factor eIF5A.Strengths:(1) The manuscript is well written, and the conclusions are, in general, appropriately cautious (besides the few improvements I suggest below).(2) The time-series inference method is interesting and promising for broader applications.(3) Simulations provide convincing support for the modeling (though some improvements are possible).(4) The reported homeostatic effect on ribosome density is surprising and carefully validated with multiple perturbations.(5) Imaging quality and corrections (e.g., flat-fielding, laser power measurements) are robust.(6) Mathematical modeling is clearly described and precise; a few clarifications could improve it further.

We thank the reviewer for recognizing the novelty of the approach and its rigour, and for providing suggestions to improve it further.

Weaknesses:(1) The absolute quantification of ribosome numbers (via the measurement of $i_{MP}$) should be improved.This only affects the finding that ribosome density is low, not that it appears to be under homeostatic control. However, if $i_{MP}$ turns out to be substantially overestimated (hence ribosome density underestimated), then "ribosomes queuing up to the initiation site and physically blocking initiation" could become a relevant hypothesis. In my detailed recommendations to the authors, I list points that need clarification in their quantifications and suggest an independent validation experiment (measuring the intensity of an object with a known number of GFP molecules, e.g., MS2-GFP MS2-GFP-labeled RNAs, or individual GEMs).

We agree with the reviewer that the estimation of the number of ribosomes is central to our finding that translation happens at low density on our reporters. This result derives from our measurement of the intensity of one mature protein (i_MP_), that we have achieved by using a SunTag reporter with a RH1 domain in the C terminus of the mature protein, allowing us to stabilise mature proteins via actin-tethering. In addition, as suggested by the reviewer, we already validated this result with an independent estimate of the mature protein intensity (Figure 5 - figure supplement 2B), which was obtained by adding the mature protein intensity directly as a free parameter of the HMM. The inferred value of mature protein intensity for each construct (10-15 a.u) was remarkably close to the experimental calibration result (14 ± 2 a.u.). Therefore, we have confidence that our absolute quantification of ribosome numbers is accurate.

(2) The proposed initiation-elongation coupling is plausible, but alternative explanations, such as changes in abortive elongation frequency, should be considered more carefully. The authors mention this possibility, but should test or rule it out quantitatively.

We thank the reviewer for the comment, but we consider that ruling out alternative explanations through new perturbation experiments is beyond the scope of the present work.

(3) The observation of translational bursting is presented as novel, but similar findings were reported by Livingston et al. (2023) using a similar SunTag-MS2 system. This prior work should be acknowledged, and the added value of the current approach clarified.

We did cite Livingston et al. (2023) in several places, but we recognized that we could add a few citations in key places, to make clear that the observation of bursting is not novel but is in agreement with previous results. We now did so in the Results and Discussion sections.

(4) It is unclear what the single-mRNA nature of the inference method is bringing since it is only used here to report _average_ ribosome elongation rate and density (averaged across mRNAs and across time during the run-off experiments - although the method, in principle, has the power to resolve these two aspects).

While decoding individual traces, our model infers shared (population-level) rates. Inferring transcript-specific parameters would be more informative, but it is highly challenging due to the uncertainty on the initial ribosome distribution on single transcripts. Pooling multiple transcripts together allows us to use some assumptions on the initial distribution and infer average elongation and initiation-rate parameters, while revealing substantial mRNA-to-mRNA variability in the posterior decoding (e.g. Figure 3 - figure Supplement 2C). Indeed, the inference still informs on the single-trace run-off time distribution (Figure 3 A) and the waiting time between termination events (Figure 3 - figure supplement 2C), suggesting the presence of stalling and bursting. In addition, the transcript-to-transcript heterogeneity is likely accounted for by our model better than previous methods (linear fit of the average run-off intensity), as suggested by their comparison (Figure 3 - figure supplement 2 A). In the future the model could be refined by introducing transcript-specific parameters, possibly in a hierarchical way, alongside shared parameters.

(5) I did not find any statement about data availability. The data should be made available. Their absence limits the ability to fully assess and reproduce the findings.

We have added the Github link in the manuscript (https://github.com/naef-lab/suntag-analysis) and have also deposited the data (.ome.tif) on Zenodo (https://zenodo.org/records/17669332).

**Recommendations for the authors:**

**Reviewer #1 (Recommendations for the authors):**
Major Comments:(1) Lack of Explicit Bursting ModelAlthough translation "bursts" are observed, the current framework does not explicitly model initiation as a stochastic ON/OFF process. This limits insight into regulatory mechanisms controlling burst frequency or duration. The authors should either incorporate a two-state/more-state (bursting) model of initiation or perform statistical analysis (e.g., dwell-time distributions) to quantify bursting dynamics. They should clarify how bursting influences the interpretation of initiation rate estimates.

We agree with the reviewer that an explicit bursting model (e.g., a two-state telegraph model) would be the ideal theoretical framework. However, integrating such a model into the TASEP-HMM inference framework is computationally intensive and complex. As a robust first step, we have opted to quantify bursting empirically based on the decoded single-mRNA traces. As shown in Figure 1G (control) and Figure 4G (perturbed conditions), we explicitly measured the duration of "ON" (translated) and "OFF" (untranslated) periods. This statistical analysis provides a quantitative description of the bursting dynamics without relying on the specific assumptions of a telegraph model. We have clarified this in the text (L123-125) and, as suggested, added a discussion (L415-417) on the potential extensions of the model to include explicit switching kinetics in the Outlook section.

(2) Assumption of Uniform Elongation RatesThe model assumes homogeneous elongation across coding sequences, which may not hold for stalling-prone inserts (e.g., PPG). This simplification could bias inference, particularly in cases of sequence-specific pausing. Adding simulations or sensitivity analysis to assess how non-uniform elongation affects the accuracy of inferred parameters. The authors should explicitly discuss how ribosome stalling, collisions, or heterogeneity might skew model outputs (see point 4).

A strong stalling sequence that affects all ribosomes equally should not deteriorate the inference of the initiation rate, provided that the low-density assumption holds. The scenario where stalling events lead to higher density, and thus increased ribosome-ribosome interactions, is comparable to the conditions explored in Figure 2E. In those simulations, we tested the inference on data generated with varying initiation and elongation rates, resulting in ribosome densities ranging from low to high. We demonstrated that the inference remains robust at low ribosome densities (<10%). At higher densities, the accuracy of the initiation rate estimate decreases, whereas the elongation rate estimate remains comparatively robust. Additionally, the model tends to overestimate ribosome density under high-density conditions, likely because it neglects ribosome interference at the initiation site (Figure 2 figure supplement 2C). We agree that a deeper investigation into the consequences of stochastic stalling and bursting would be beneficial, and we have explicitly acknowledged this in the Limitations section.

(3) Interpretation of eIF5A Knockout PhenotypeThe observation that eIF5A KO reduces initiation more than elongation, leading to decreased ribosome density, is biologically intriguing. However, the explanation invoking altered RQC kinetics is speculative and not directly tested. The authors should consider validating the RQC hypothesis by monitoring reporter mRNA stability, ribosome collision markers, or translation termination intermediates.

We thank the reviewer for the comment, but we consider that ruling out alternative explanations through new experiments is beyond the scope of the present work.

(4) To strengthen the manuscript, the authors should incorporate insights from three studies.- Livingston et al. (PMC10330622) found that translation occurs in bursts, influenced by mRNA features and initiation factors, supporting the coupling of initiation and elongation.- Madern et al. (PMID: 39892379) demonstrated that ribosome cooperativity enhances translational efficiency, highlighting coordinated ribosome behavior.- Dufourt et al. (PMID: 33927056) observed that high initiation rates correlate with high elongation rates, suggesting a conserved mechanism across cell cultures and organisms.Integrating these studies could enrich the manuscript's interpretation and stimulate new avenues of thought.

We thank the reviewer for the valuable comment. We added citations of Livingston et al. in the context of translational bursting. We already cited Madern et al. in multiple places and, although its observations of ribosome cooperativity are very compelling, they cannot be linked with our observations of a feedback between initiation and elongation, and it would be very challenging to see a similar effect on our reporters. This is why we did not expressly discuss cooperativity. We also integrated Dufourt et al. in the Discussion about the possibility of designing genetically-encoded reporter. We also added a sentence about the possibility of using an ER-specific SunTag reporter, as done recently in Choi et al., Nature (2025) (https://doi.org/10.1038/s41586-025-09718-0).

Minor Comments:(1) Use consistent naming for SunTag reporters (e.g., "PPG" vs "proline-rich") throughout.

Thank you for the comment. However, the term proline-rich always appears together with PPG, so we believe that the naming is clear and consistent.

(2) Consider a schematic overview of the experimental design and modeling pipeline for accessibility.

Thank you for the suggestion. We consider that experimental design and modeling is now sufficiently clearly described and does not justify an additional scheme.

(3) Clarify how incomplete run-off traces are handled in the HMM inference.

Incomplete run-off traces are treated identically to complete traces in our HMM inference. This is possible because our model relies on the probability of transitions occurring per time step to infer rates. It does not require observing the final "empty" state to estimate the kinetic parameters ɑ and λ. The loss of signal (e.g., mRNA moving out of the focal volume or photobleaching) does not invalidate the kinetic information contained in the portion of the trace that was observed. We have clarified this in the Methods section.

**Reviewer #2 (Recommendations for the authors):**
(1) Reproducibility:(1.1) The authors should use a GitHub repository with a timestamp for the release version.

The code is available on GitHub (https://github.com/naef-lab/suntag-analysis).

(1.2) Make raw images and data available in a figure repository like Figshare.

The raw images (.ome.tif) are now available on Zenodo (https://zenodo.org/records/17669332).

(2) Paper reorganization and expansion of the intensity and ribosome quantification:(2.1) Given the relevance of the initiation and elongation rates for the conclusions of this study, and the fact that the authors inferred these rates from the spot intensities. I recommend that the authors move Figure 1 Supplement 2 to the main text and expand the description of the process to relate spot intensity and number of ribosomes. Please also expand the figure caption for this image.

We agree with the importance of this validation. We have expanded the description of the calibration experiment in the main text and in the figure caption.

(2.2) I suggest the authors explicitly mention the use of HMM in the abstract.

We have now explicitly mentioned the TASEP-based HMM in the abstract.

(2.3) In line 492, please add the frame rate used to acquire the images for the run-off assays.

We have added the specific frame rate (one frame every 20 seconds) to the relevant section.

(3) Figures and captions:(3.1) Figure 1, Supplement 2. Please add a description of the colors used in plots B, C.

We have expanded the caption and added the color description.

(3.2) In the Figure 2 caption. It is not clear what the authors mean by "traceseLife". Please ensure it is not a typo.

Thank you for spotting this. We have corrected the typo.

(3.3) Figure 1 A, in the cartoon N(alpha)->N-1, shouldn't the transition also depend on lambda?

The transition probability was explicitly derived in the “Bayesian modeling of run-off traces” section (Eqs. 17-18), and does not depend on λ, but only on the initiation rate under the low-density assumption.

(3.4) Figure 3, Supplement 2. "presence of bursting and stalling.." has a typo.

Corrected.

(3.5) Figure 5, panel C, the y-axis label should be "run-off time (min)."

Corrected.

(3.6) For most figures, add significance bars.(3.7) In the figure captions, please add the total number of cells used for each condition.

We have systematically indicated the number of traces (n_t_) and the number of independent experiments (n_e_) in the captions in this format (n_t_, n_e_).

(4) Mathematical Methods:

We greatly thank the reviewer for their detailed attention to the mathematical notation. We have addressed all points below.

(4.1) In lines 555, Materials and Methods, subsection, Quantification of Intensity Traces, multiple equations are not numbered. For example, after Equation (4), no numbers are provided for the rest of the equations. Please keep consistency throughout the whole document.

We have ensured that all equations are now consistently numbered throughout the document.

(4.2) In line 588, the authors mention "$X$ is a standard normal random variable with mean $\mu$ and standard deviation $s_0$". Please ensure this is correct. A standard normal random variable has a 0 mean and std 1.

Thank you for the suggestion, we have corrected the text (L678).

(4.3) Line 546, Equation 2. The authors use mu(x,y) to describe a 2d Gaussian function. But later in line 587, the authors reuse the same variable name in equation 5 to redefine the intensity as mu = b_0 + I.

We have renamed the 2D Gaussian function to \mu_{2D}(x,y) in the spot tracking section

(4.4) For the complete document, it could be beneficial to the reader if the authors expand the definition of the relationship between the signal "y" and the spot intensity "I". Please note how the paragraph in lines 582-587 does not properly introduce "y".

We have added an explicit definition of y and its relationship to the underlying spot intensity I in the text to improve readability and clarity.

(4.5) Please ensure consistency in variable names. For example, "I" is used in line 587 for the experimental spot intensity, then line 763 redefines I(t) as the total intensity obtained from the TASEP model; please use "I_sim(t)" for simulated intensities. Please note that reusing the variable "I" for different contexts makes it hard for the reader to follow the text.

We agree that this was confusing. We have implemented the suggestion and now distinguish simulated intensities using the notation I_S_ .

(4.6) Line 555 "The prior on the total intensity I is an "uninformative" prior" I ~ half_normal(1000). Please ensure it is not "I_0 ~ half_normal(1000)."?

We confirm that “I” is the correct variable representing the total intensity in this context; we do not use an “I_0_” variable here.

(4.7) In lines 595, equation 6. Ensure that the equation is correct. Shouldn't it be: s_0^2 = ln (1+(sigma_meas^2/⟨y⟩^2))? Please ensure that this is correct and it is not affecting the calculated values given in lines 598.

Thank you for catching this typo. We have corrected the equation in the manuscript. We confirm that the calculations performed in the code used the correct formula, so the reported values remain unchanged.

(4.8) In line 597, "the mean intensity square ^2". Please ensure it is not "the square of the temporal mean intensity."

We have corrected the text to "the square of the temporal mean intensity."

(4.9) In lines 602-619, Bayesian modeling of run-off traces, please ensure to introduce the constant "\ell". Used to define the ribosomal footprint?

We have added the explicit definition of 𝓁 as the ribosome footprint size (length of transcript occupied by one ribosome) in the "Bayesian modeling of run-off traces" section.

(4.10) Line 687 has a minor typo "[...] ribosome distribution.. Then, [...]"

We have corrected the punctuation.

(4.11) In line 678, Equation 19 introduces the constant "L_S", Please ensure that it is defined in the text.

We have added the explicit definition of L_S_ (the length of the SunTag) to the text surrounding Equation 19.

(4.12) In line 695, Equation 22, please consider using a subscript to differentiate the variance due to ribosome configuration. For example, instead of "sigma (...)^2" use something like "sigma_c ^2 (...)". Ensure that this change is correctly applied to Equation 24 and all other affected equations.

Thank you, we have implemented the suggestions.

(4.13) In line 696, please double-check equations 26 and 27. Specifically, the denominator ^2. Given the previous text, it is hard to follow the meaning of this variable.

We have revised the notation in Equations 26 and 27 to ensure the denominator is consistent with the definitions provided in the text.

(4.14) In lines 726, the authors mention "[...], but for the purposes of this dissertation [...]", it should be "[...], but for the purposes of this study [...]"

Thank you for spotting this. We have replaced "dissertation" with "study."

(4.15) Equations 5, 28, 37, and the unnumbered equation between Equations 16 and 17 are similar, but in some, "y" does not explicitly depend on time. Please ensure this is correct.

We have verified these equations and believe they are correct.

(4.16) Please review the complete document and ensure that variables and constants used in the equations are defined in the text. Please ensure that the same variable names are not reused for different concepts. To improve readability and flow in the text, please review the complete Materials and Methods sections and evaluate if the modeling section can be written more clearly and concisely. For example, Equation 28 is repeated in the text.

We have performed a comprehensive review of the Materials and Methods section. To improve conciseness and flow, we have merged the subsection “Observation model and estimation of observation parameters” with the “Bayesian modeling of run-off traces” section. This allowed us to remove redundant definitions and repeated equations (such as the previous Equation 28). We have also checked that all variables and constants are defined upon first use and that variable names remain consistent throughout the manuscript.

**Reviewer #3 (Recommendations for the authors):**
(1) Data Presentation(1.1) In main Figures 1D and 4E, the traces appear to show frequent on-off-on transitions ("bursting"), but in supplementary figures (1-S1A and 4-S1A), this behavior is seen in only ~8 of 54 traces. Are the main figure examples truly representative?

We acknowledge the reviewer's point. In Figure 1D, we selected some of the longest and most illustrative traces to highlight the bursting dynamics. We agree that the term "representative" might be misleading if interpreted as "average." We have updated the text to state "we show bursting traces" to more accurately reflect the selection.

(1.2) There are 8 videos, but I could not identify which is which.

Thank you for pointing this out. We have renamed the video files to clearly correspond to the figures and conditions they represent.

(2) Data Availability:As noted above, the data should be shared. This is in accordance with eLife's policy: "Authors must make all original data used to support the claims of the paper, or that are required to reproduce them, available in the manuscript text, tables, figures or supplementary materials, or at a trusted digital repository (the latter is recommended). [...] eLife considers works to be published when they are posted as preprints, and expects preprints we review to meet the standards outlined here." Access to the time traces would have been helpful for reviewers.

We have now added the Github link for the code (https://github.com/naef-lab/suntag-analysis) and deposited the raw data (.ome.tif files) on Zenodo (10.5281/zenodo.17669332).

(3) Model Assumptions:(3.1) The broad range of run-off times (Figure 3A) suggests stalling, which may be incompatible with the 'low-density' assumption used on the TASEP model, which essentially assumes that ribosomes do not bump into each other. This could impact the validity of the assumptions that ribosomes behave independently, elongate at constant speed (necessary for the continuum-limit approximation), and that the rate-limiting step is the initiation. How robust are the inferences to this assumption?

We agree that the deviation of waiting times from an exponential distribution (Figure 3 - figure supplement 2C) suggests the presence of stalling, which challenges the strict low-density assumption and constant elongation speed. We explicitly explored the robustness of our model to higher ribosome densities in simulations. As shown in Figure 2 - figure supplement 2, while the model accuracy for single parameters deteriorates at very high densities (overestimating density due to neglected interference), it remains robust for estimating global rates in the regime relevant to our data. We have expanded the discussion on the limitations of the low density and homogeneous elongation rate assumptions in the text (L404-408).

(3.2) Since all constructs share the same SunTag region, elongation rates should be identical there and diverge only in the variable region. This would affect $\gamma (t)$ and hence possibly affect the results. A brief discussion would be helpful.

This is a valid point. Currently, our model infers a single average elongation rate that effectively averages the behavior over the SunTag and the variable CDS regions. Modeling distinct rates for these regions would be a valuable extension but adds significant complexity. While our current "effective rate" approach might underestimate the magnitude of differences between reporters, it captures the global kinetic trend. We have added a brief discussion acknowledging this simplification (L408-412).

(3.3) A similar point applies to the Gillespie simulations: modeling the SunTag region with a shared elongation rate would be more accurate.

We agree. Simulating distinct rates for the SunTag and CDS would increase realism, though our current homogeneous simulations serve primarily to benchmark the inference framework itself. We have noted this as a potential future improvement (L413-414).

(3.4) Equation (13) assumes that switching between bursting and non-bursting states is much slower than the elongation time. First, this should be made explicit. Second, this is not quite true (~5 min elongation time on Figure 3-s2A vs ~5-15min switching times on Figure 1). It would be useful to show the intensity distribution at t=0 and compare it to the expected mixture distribution (i.e., a Poisson distribution + some extra 'N=0' cells).

We thank the reviewer for this insightful comment. We have added a sentence to the text explicitly stating the assumption that switching dynamics are slower than the translation time. While the timescales are indeed closer than ideal (5 min vs. 5-15 min), this assumption allows for a tractable approximation of the initial conditions for the run-off inference. Comparing the intensity distribution at t=0 to a zero-inflated Poisson distribution is an excellent suggestion for validation, which we will consider for future iterations of the model.

(4) Microscopy Quantifications:(4.1) Figure 1-S2A shows variable scFv-GFP expression across cells. Were cells selected for uniform expression in the analysis? Or is the SunTag assumed saturated? which would then need to be demonstrated.

All cell lines used are monoclonal, and cells were selected via FACS for consistent average cytoplasmic GFP signal. We assume the SunTag is saturated based on the established characterization of the system by Tanenbaum et al. (2014), where the high affinity of the scFv-GFP ensures saturation at expression levels similar to ours.

(4.2) As translation proceeds, free scFv-GFP may become limiting due to the accumulation of mature SunTag-containing proteins. This would be difficult to detect (since mature proteins stay in the cytoplasm) and could affect intensity measurements (newly synthesized SunTag proteins getting dimmer over time).

This effect can occur with very long induction times. To mitigate this, we optimized the Doxycycline (Dox) incubation time for our harringtonine experiments to prevent excessive accumulation of mature protein. We also monitor the cytoplasmic background for granularity, which would indicate aggregation or accumulation.

(4.3) The statements "for some traces, the mRNA signal was lost before the run-off completion" (line 195) and "we observed relatively consistent fractions of translated transcripts and trace duration distributions across reporters" (line 340) should be supported by a supplementary figure.

The first statement is supported by Figure 2 - figure supplement 1, which shows representative run-off traces for all constructs, including incomplete ones.

The second statement regarding consistency is supported by the quantitative data in Figure 1E and G, which summarize the fraction of translated transcripts and trace durations across conditions.

(4.4) Measurements of single mature protein intensity $i_{MP}$:(4.4.1) Since puromycin is used to disassemble elongating ribosomes, calibration may be biased by incomplete translation products (likely a substantial fraction, since the Dox induction is only 20min and RNAs need several minutes to be transcribed, exported, and then fully translated).

As mentioned in the “Live-cell imaging” paragraph, the imaging takes place 40 min after the end of Dox incubation. This provides ample time for mRNA export and full translation of the synthesized proteins. Consequently, the fraction of incomplete products generated by the final puromycin addition is negligible compared to the pool of fully synthesized mature proteins accumulated during the preceding hour.

(4.4.2) Line 519: "The intensity of each spot is averaged over the 100 frames". Do I understand correctly that you are looking at immobile proteins? What immobilizes these proteins? Are these small aggregates? It would be surprising that these aggregates have really only 1, 2, or 3 proteins, as suggested by Figure 1-S2A.

We are visualizing mature proteins that are specifically tethered to the actin cytoskeleton. This is achieved using a reporter where the RH1 domain is fused directly to the C-terminus of the Renilla protein (SunTag-Renilla-RH1). The RH1 domain recruits the endogenous Myosin Va motor, which anchors the protein to actin filaments, rendering it immobile. Since each Myosin Va motor interacts with one RH1 domain (and thus one mature protein), the resulting spots represent individual immobilized proteins rather than aggregates. We have now revised the text and Methods section to make this calibration strategy and the construct design clearer (L130-140).

(4.4.3) Estimating the average intensity $i_{MP}$ of single proteins all resides in the seeing discrete modes in the histogram of Figure 1-S2B, which is not very convincing. A complementary experiment, measuring *on the same microscope* the intensity of an object with a known number of GFP molecules (e.g., MS2-GFP labeled RNAs, or individual GEMs https://doi.org/10.1016/j.cell.2018.05.042 (only requiring a single transfection)) would be reassuring to convince the reader that we are not off by an order of magnitude.

While a complementary calibration experiment would be valuable, we believe our current estimate is robust because it is independently validated by our model. When we inferred i_MP_ as a free parameter in the HMM (Figure 5 - figure supplement 2B), the resulting value (10-15 a.u.) was remarkably consistent with our experimental calibration (14 ± 2 a.u.). We have clarified this independent validation in the text to strengthen the confidence in our quantification (L264-272).

(4.4.4) Further on the histogram in Figure 1-S2B:- The gap between the first two modes is unexpectedly sharp. Can you double-check? It means that we have a completely empty bin between two of the most populated bins.

We have double-checked the data; the plot is correct, though the sharp gap is likely due to the small sample size (n=29).

- I am surprised not to see 3 modes or more, given that panel A shows three levels of intensity (the three colors of the arrows).

As noted below, brighter foci exist but fall outside the displayed range of the histogram.

- It is unclear what the statistical test is and what it is supposed to demonstrate.

The Student's t-test compares the means of the two identified populations to confirm they are statistically distinct intensity groups.

- I count n = 29, not 31. (The sample is small enough that the bars of the histogram show clear discrete heights, proportional to 1, 2, 3, 4, and 5 --adding up all the counts, I get 29). Is there a mistake somewhere? Or are some points falling outside of the displayed x-range?

You are correct. Two brighter data points fell outside the displayed range. The total number of foci in the histogram is 29. We have corrected the figure caption and the text accordingly.

(5) Miscellaneous Points:(5.1) Panel B in Figure 2-s1 appears to be missing.

The figure contains only one panel.

(5.2) In Equation (7), $l$ is not defined (presumably ribosome footprint length?). Instead, $J$ is defined right after eq (7), as if it were used in this equation.

Thank you for pointing this out, we have corrected it.

(5.3) Line 703, did you mean to write something else than "Equation 26" (since equation 26 is defined after)?

Yes, this was a typo. We have corrected the cross-reference.